# Realizing efficient blue and deep-blue delayed fluorescence materials with record-beating electroluminescence efficiencies of 43.4%

Yan Fu[1], Hao Liu [1], Ben Zhong Tang[2] & Zujin Zhao [1] ✉

As promising luminescent materials for organic light-emitting diodes (OLEDs), thermally activated delayed fluorescence materials are booming vigorously in recent years, but robust blue ones still remain challenging. Herein, we report three highly efficient blue and deep-blue delayed fluorescence materials comprised of a weak electron acceptor chromeno[3,2-c]carbazol-8(5H)-one with a rigid polycyclic structure and a weak electron donor spiro[acridine-9,9′-xanthene]. They hold distinguished merits of excellent photoluminescence quantum yields (99%), ultrahigh horizontal transition dipole ratios (93.6%), and fast radiative transition and reverse intersystem crossing, which furnish superb blue and deep-blue electroluminescence with Commission Internationale de l'Eclairage coordinates ($CIE_{x,y}$) of (0.14, 0.18) and (0.14, 0.15) and record-beating external quantum efficiencies ($\eta_{ext}$s) of 43.4% and 41.3%, respectively. Their efficiency roll-offs are successfully reduced by suppressing triplet-triplet and singlet-singlet annihilations. Moreover, high-performance deep-blue and green hyperfluorescence OLEDs are achieved by utilizing these materials as sensitizers for multi-resonance delayed fluorescence dopants, providing state-of-the-art $\eta_{ext}$s of 32.5% ($CIE_{x,y}$ = 0.14, 0.10) and 37.6% ($CIE_{x,y}$ = 0.32, 0.64), respectively, as well as greatly advanced operational lifetimes. These splendid results can surely inspire the development of blue and deep-blue luminescent materials and devices.

Organic light-emitting diodes (OLEDs) have achieved tremendous advancements in recent years, and have become a highly promising technique for display and illumination[1–3]. The luminescent materials are the essential components for OLEDs, and have experienced the development from organic fluorophores to organometallic phosphors with exciton utilization enhanced from 25% to 100%[4]. The emergence of purely organic thermally activated delayed fluorescence (TADF) materials that can theoretically utilize 100% electrogenerated excitons

without the assistance of heavy metals provides a new breakthrough for OLEDs[3,5,6]. Through efficient reverse intersystem crossing (RISC) process, the triplet excitons can be up-converted to singlet excitons, endowing TADF materials with high electroluminescence (EL) efficiencies. This RISC process is mainly driven by small energy splitting ($\Delta E_{ST}$) between the lowest excited singlet ($S_1$) and triplet ($T_1$) states. TADF materials are usually designed into highly twisted electron donor (D)-acceptor (A) structures for effectively separating the highest

[1]State Key Laboratory of Luminescent Materials and Devices, Guangdong Provincial Key Laboratory of Luminescence from Molecular Aggregates, South China University of Technology, Guangzhou 510640, China. [2]School of Science and Engineering, Shenzhen Institute of Aggregate Science and Technology, The Chinese University of Hong Kong, Shenzhen, Guangdong 518172, China. ✉e-mail: mszjzhao@scut.edu.cn

occupied molecular orbitals (HOMOs) and the lowest unoccupied molecular orbitals (LUMOs) to acquire small $\Delta E_{ST}$s and thus fast RISC process. Besides, strong spin-orbital coupling (SOC) between the excited states with different spin multiplicities and transition properties also facilitates RISC process[7–9].

To date, efficient red, green and sky-blue TADF materials have been reported, while the exploration of blue and deep-blue ones with high EL efficiencies remains challenging, because the strong charge transfer (CT) effect and large vibrational relaxation on account of the highly twisted D-A structures often lead to broad and long-wavelength emissions[10–15]. And the knotty issues of serious efficiency roll-off and poor operational stability have to be addressed for blue and deep-blue TADF materials as well. The triplet-triplet annihilation (TTA) and triplet-polaron annihilation (TPA) are widely recognized as the main cause for EL efficiency loss at high voltages for most TADF materials, which are attributed to the insufficient RISC rate ($k_{RISC}$) and unbalanced carrier transport, respectively[12–19]. Moreover, due to the CT-dominated transition, the radiative decay rates ($k_r$s) of TADF materials are always 1–2 orders of magnitude smaller than those of fluorescent materials with locally excited (LE) dominated transition. Hence, the singlet-singlet annihilation (SSA) could be another exciton loss mechanism[18,20,21]. These considerations suggest that a rigid molecular structure with weak D-A interaction and fast RISC process and radiative transition (i.e. large $k_{RISC}$ and $k_r$) are important for creating robust blue and deep-blue TADF materials.

In this work, we wish to report the design and preparation of a series of highly efficient blue and deep-blue TADF materials comprised of a weak and rigid electron donor spiro[acridine-9,9'-xanthene] (SXAC) and a carbonyl-containing weak electron acceptor 5-phenyl-chromeno[3,2-c]carbazol-8(5H)-one (CCO-α) that has a rigid polycyclic structure. CCO-α can be regarded as a ring-closed structure of phenyl(9-phenyl-9H-carbazol-3-yl)methanone (CBP-α) by an oxygen bridge, which has higher rigidity and planarity, better conjugation and

lower electron-withdrawing ability. The molecular frameworks built with SXAC and CCO-α can ensure weak D-A interaction and reduced CT effect, and bulky *tert*-butyl group is introduced to modulate intermolecular distance and interaction (Fig. 1). These tailor-made new materials (CCO-1, CCO-2 and CCO-3) show strong blue delayed fluorescence with large $k_r$ and $k_{RISC}$, and excellent photoluminescence (PL) quantum yields ($\Phi_{PL}$s) reaching 99%. Meanwhile, the enhanced planarity by CCO-α brings about high horizontal transition dipole ratios ($\Theta_{//}$s) of up to 93.6% and thus brilliant light out-coupling efficiencies ($\eta_{out}$s) of up to 43.9%[15,22,23]. Highly efficient blue and deep-blue OLEDs are achieved, affording record-beating external quantum efficiencies ($\eta_{ext}$s) of 43.4% and 41.3% with Commission Internationale de I'Eclairage coordinates ($CIE_{x,y}$) of (0.14, 0.18) and (0.14, 0.15), respectively. Furthermore, high-performance deep-blue and green hyperfluorescence (HF) OLEDs with outstanding $\eta_{ext}$ of 32.5% ($CIE_{x,y} = 0.14$, 0.10) and 37.6% ($CIE_{x,y} = 0.32$, 0.64) and improved operational lifetimes are also realized.

## Results

### Molecular design concept, thermal stability and crystallographic property

The design of rigid electron acceptor with weak electron-withdrawing ability is critical in this work. According to previous reports, electron-withdrawing benzoyl is favourable for the construction of delayed fluorescence materials due to the intrinsic n–π* transition of its carbonyl, which exert positive impacts over strengthening SOC effect and facilitating RISC process[15,16,24,25]. In addition, direct attaching electron-rich carbazole to the carbonyl in benzoyl can weaken its electron-withdrawing ability, which is conducive to building efficient delayed fluorescence materials with blue-shifted emission by introducing another electron donor on benzoyl via its phenyl bridge[15]. But the carbazole-benzoyl framework is relatively flexible, which is not perfect in view of reducing nonradiative decay. Therefore, a rigid polycyclic

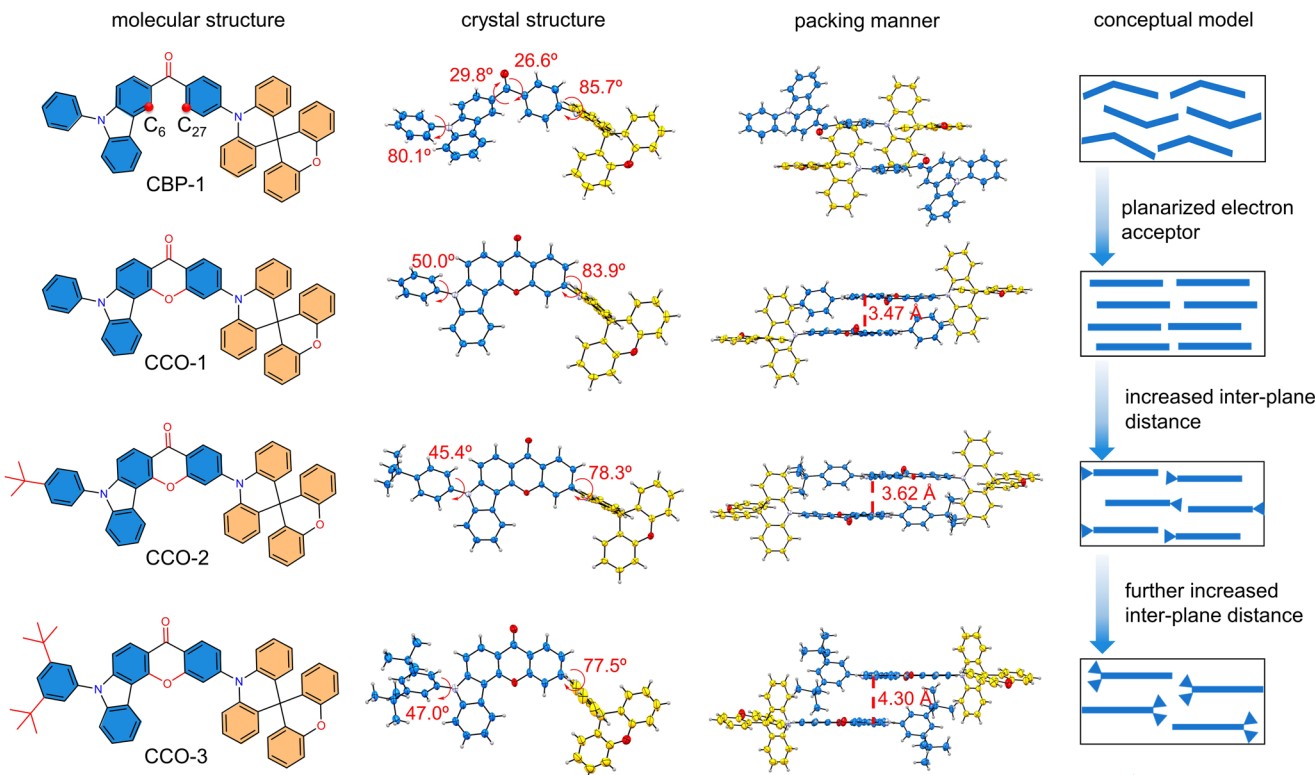

**Fig. 1 | Design strategy for high-efficiency blue delayed fluorescence materials.** Molecular structures, crystal structures, packing manners and conceptual models of CBP-1, CCO-1, CCO-2, and CCO-3. The dihedral angles are labeled on the crystal structures. The red dashed lines in the third column refer to the interplane distances between two acceptors with distance values in red.

structure with lowered electron-withdrawing ability is envisioned by ring closure with an oxygen linkage in carbazole-benzoyl framework. The selection of oxygen linkage is because its lone pair of electrons can delocalize to carbonyl to further weaken the electron-withdrawing ability. The positions for ring-closure are analyzed by relaxed scan on the control material (CBP-1), performed by adjusting the dihedral angle between carbazole and carbonyl step by step (Supplementary Fig. 1). The results suggest that a stable molecular configuration with the lowest energy can be achieved by ring-closure at the $C_6$ and $C_{27}$ positions as illustrated in Fig. 1. The carbonyl-containing polycyclic electron acceptor CCO-$\alpha$, target materials (CCO-1, CCO-2 and CCO-3) and control material (CBP-1) are prepared according to the synthetic routes outlined in Supplementary Fig. 2. The detailed procedures and characterization data are described in Supplementary Materials. All the final materials are thermally stable with high decomposition temperatures over 450 °C at 5 wt% initial weight loss and no clear glass-transition temperature (Supplementary Fig. 4). The single crystal structures unveil that SXAC donor is connected to CCO-$\alpha$ acceptor in a twisted conformation with large dihedral angles of 77.5–83.9°. CCO-$\alpha$ has a planar structure, and adopts a face-to-face stacking manner in crystals, which facilitates carrier transport[26,27]. The inter-plane distance between two adjacent CCO-$\alpha$ moieties is increased from 3.47 Å of CCO-1 to 3.62 Å of CCO-2 and to 4.30 Å of CCO-3 (Fig. 1 and Supplementary Figs. 3, 5), indicating tert-butyl groups have effectively modulated the intermolecular π–π interaction of CCO-$\alpha$.

## Theoretical calculation and analysis

To investigate the influence of ring-closed structure, the reorganization energy ($\lambda$) and Huang-Rhys factor ($S$) between $S_0$ and $S_1$ states are calculated for these materials, which can quantify the displacements of the nuclear coordinates ($D$)[28]. The sum of $\lambda$ and $S$ for the normal mode of $S_0/S_1$ states are reduced from 0.253/0.298 eV and 23.80/34.36 of CBP-1 to 0.187/0.183 eV and 4.14/4.28 of CCO-1. And the root mean squared displacement (RMSD) between $S_0$ and $S_1$ states is decreased from 0.557 Å of CBP-1 to 0.230 Å of CCO-1. By decomposing the total $\lambda$ and $S$ into the contribution of each vibration mode, it is found that the vibrations of carbazole-benzoyl framework are mainly responsible for conformation change in CBP-1 (Fig. 2 and Supplementary Fig. 6). But in CCO-1, the twisting vibrations at low frequencies (<200 cm$^{-1}$) and the stretching and scissoring vibrations at high frequencies (>1000 cm$^{-1}$) of benzoyl and carbazole are greatly restricted, indicative of significantly improved rigidity by ring-closure (Supplementary Figs. 7–10). The atomic dipole moment corrected Hirshfeld (ADCH) charge population analysis[29] discloses that the negative charge on the oxygen atom of carbonyl is increased from −0.305 in CBP-1 to −0.319 in CCO-1, while the positive charge on the carbon atom of carbonyl is decreased from +0.229 in CBP-1 to +0.174 in CCO-1, disclosing the electron-withdrawing ability of CCO-$\alpha$ is lowered intrinsically. However, with the inductive effect[30,31] of the oxygen bridge in CCO-$\alpha$, the LUMO energy level ($E_{LUMO}$) of CCO-$\alpha$ is lowered than that of CBP-$\alpha$ (Supplementary Fig. 11). While the energy level of HOMO (LUMO) of a D-A-type molecule is driven in a first approximation by both the relative energy and the electronic interaction between the HOMOs (LUMOs) of the electron donor (D) and acceptor (A) fragments, and the electronic interaction of CCO-1 and CBP-1 is similar due to the close dihedral angles between D and A fragments (Supplementary Fig. 12). Hence, the lower relative energy of CCO-$\alpha$ with respect to CBP-$\alpha$ results in a lower $E_{LUMO}$ of CCO-1. An identical reasoning can be adopted for the evolution of HOMO energy level ($E_{HOMO}$) of CBP-1 and CCO-1. However, due to the small orbital coefficients in HOMO at the $C_6$ and $C_{26}$ positions of CBP-1 (Supplementary Fig. 11), the influence of the inductive effect on $E_{HOMO}$ is weaker in comparison with that on $E_{HOMO}$. Besides, the $E_{HOMO}$ of CCO-$\alpha$ is further destabilized by the mesomeric effect of the oxygen bridge, due to the low-lying energy level of the oxygen's lone pair in relation to this orbital[30]. Thus, the decrease in $E_{HOMO}$ of CCO-$\alpha$ is

smaller than that of $E_{LUMO}$. Overall, the HOMO-LUMO gap of CCO-1 (3.68 eV) is decreased compared with that of CBP-1 (3.69 eV), but the difference between them is actually neglectable, which will avoid the red-shifted emission because of the expansion of conjugation. Concerning the smaller $S$ and $\lambda$ resulting from the increased molecular rigidity and weakened D-A interaction, CCO-1 is anticipated to achieve stronger and bluer emission with a narrower FWHM than CBP-1.

The natural transition orbital (NTO) analysis performed on the excited states uncover that, for $S_1$ state, the highest occupied natural transition orbital (HONTO) is concentrated on SXAC, while the lowest unoccupied natural transition orbital (LUNTO) is mainly distributed on carbonyl and two adjacent benzenes (Fig. 2b and Supplementary Fig. 13). Due to weakened D-A interaction and thus lowered CT effect of $S_1$ state, the LE feature of CCO-1 in $S_1$ state is enhanced accordingly, accompanied by a larger overlap integral of norm of HONTO and LUNTO ($OI_{H/L}$) of 0.132 than that of CBP-1 (0.117), which is conducive to radiative transition. The calculated $T_1$ energy level of CBP-1 is much lower than that of CCO-1, and thus the theoretical $\Delta E_{st}$ of CBP-1 (0.17 eV) becomes larger than that of CCO-1 (0.08 eV). The lower $T_1$ energy level of CBP-1 discloses a larger exchange integral ($K$), which can endow the $T_1$ state of CBP-1 with increased LE component[9]. The $OI_{H/L}$ for the $T_1$ state of CBP-1 is calculated to be 0.615, larger than that of CCO-1 (0.486), which also validates a higher LE component of the $T_1$ state. Due to more different transition characteristics between $S_1$ and $T_1$ states, the SOC constant of CBP-1 (0.77 cm$^{-1}$) is larger than that of CCO-1 (0.36 cm$^{-1}$), suggesting the bridging oxygen atom does not contribute to SOC. But the ring-closed structure by oxygen atom can influence excited-state energy levels, leading to a smaller $\Delta E_{st}$ for CCO-1. So, the $k_{RISC}$ from $T_1$ to $S_1$ state of CCO-1 is calculated to be 2.46 × 10$^6$ s$^{-1}$, larger than that of CBP-1 (9.14 × 10$^5$ s$^{-1}$), despite its smaller SOC constant. Moreover, the $T_2$ energy level of CCO-1 lies close and slightly higher to its $S_1$ energy level, and the gap between $T_1$ and $T_2$ energy levels is only 0.12 eV, small enough for vibronic coupling to induce SOC interaction between $T_2$ and $S_1$ states (Fig. 2c and Supplementary Table 1). Therefore, $T_1$ and $T_2$ wavefunctions can be mixed by vibronic coupling, opening supplementary RISC channel from $T_2$ to $S_1$ states in CCO-1[7,9]. However, the $T_2$ energy level of CBP-1 (2.53 eV) is almost equal to $T_1$ energy level (2.51 eV), which makes it possible to form a conical intersection between $T_2$ and $T_1$ states. This virtually will provoke consecutive and fast internal conversion (IC) process to compete with RISC process[32,33]. Therefore, in comparison with CBP-1, CCO-1 is more likely to have a faster RISC process with the aid of its smaller $\Delta E_{st}$ and double RISC channels. CCO-2 and CCO-3 have similar theoretical results to CCO-1 (Supplementary Figs. 13–15, and Supplementary Table 1).

The transition dipole moments (TDMs) of CCO-1 and CBP-1 are calculated in solid state based on crystal structures. The two-layer ONIOM approach combining quantum mechanics and molecular mechanics (QM/MM) is employed in calculation, where the central molecule is defined as the active QM part, and the surrounding molecules act as MM part. According to Born-Oppenheimer approximation, the TDM is influenced by nuclear vibration configuration, electron spin configuration and electron orbital wavefunction, and the large overlap of transition orbitals is favorable for gaining high value of TDM ($\mu$). Besides, oscillator strength ($f$) is related to $\mu$ according to Eq. (1)[34]:

$$f = \frac{2m_e \Delta E}{3\hbar^2 e^2} \mu^2 \tag{1}$$

where $m_e$ is the mass of electron, $\Delta E$ is the energy of transition, $\hbar$ is the reduced Planck's constant, and $e$ is the charge of electron. As excepted, CCO-1 has a larger $\mu$ than CBP-1, which brings about a larger $f$ value, and the TDM direction of CCO-1 dominantly aligns along the $x$-$y$ plane (Fig. 2d and Supplementary Table 2). During evaporation, CCO-1 has a

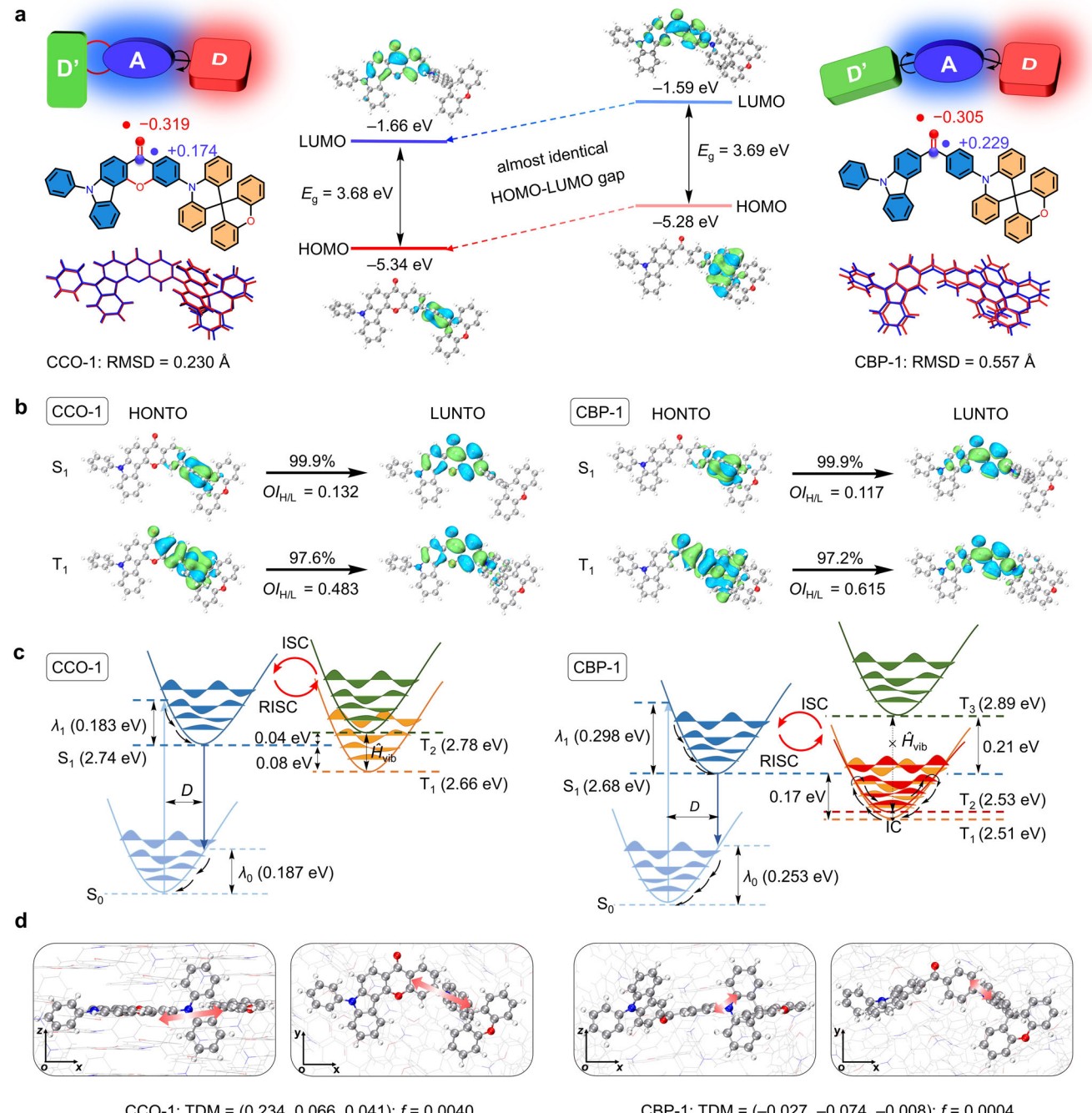

**Fig. 2 | Analyses of quantum chemical calculations results for CCO-1 and CBP-1.** **a** Calculated orbital distributions and energy levels of HOMO and LUMO, atomic dipole moment corrected Hirshfeld charge populations of carbonyl, and the root mean squared displacement (RMSD) between optimized $S_0$ (red) and $S_1$ (blue) geometries. **b** NTO distribution and overlap integral of norm of HONTO and LUNTO ($OI_{H/L}$) of $S_1$ and $T_1$ states for CCO-1 and CBP-1, respectively. **c** Energy level and photophysical processes among $S_0$, $S_1$, $T_1$, and $T_2$ states. $\lambda_0$ and $\lambda_1$ refer to reorganization energy for $S_0$ and $S_1$ states. D refers to displacements of the nuclear coordinates. $\hat{H}_{vib}$, IC, ISC and RISC refer to the non-adiabatic vibronic coupling, internal crossing, intersystem crossing, and reverse intersystem crossing processes. **d** Transition dipole moment (TDM) and the corresponding oscillator strength ($f$) values in solid state.

higher tendency to align horizontally on substrate with a larger $\Theta_{//}$ than CBP-1. These findings also validate the presence of planar polycyclic CCO-$\alpha$ has positive effect to promote horizontal transition dipole orientation. Moreover, according to Einstein's spontaneous emission Eq. (2)[32]:

$$k_r = \frac{8\pi^2 \triangle E^3}{3\varepsilon_0 \hbar c^3}\mu^2 \approx \frac{f \triangle E^2}{1.5} \qquad (2)$$

where $\varepsilon_0$ is the permittivity of free space, and $k_r$ is positively related to $f$ value. Benefiting from a much larger $f$ value, the $k_r$ of CCO-1 calculated in solid state is $1.2 \times 10^6 \, s^{-1}$, being about ten-fold larger than that of CBP-1 ($1.1 \times 10^5 \, s^{-1}$), indicative of faster radiative transition. Therefore, the SSA process can be better alleviated in CCO-1.

**Photophysical property**

In toluene solutions, CCO-1, CCO-2 and CCO-3 show almost identical absorption profiles with absorption maxima at 364 nm, associated with

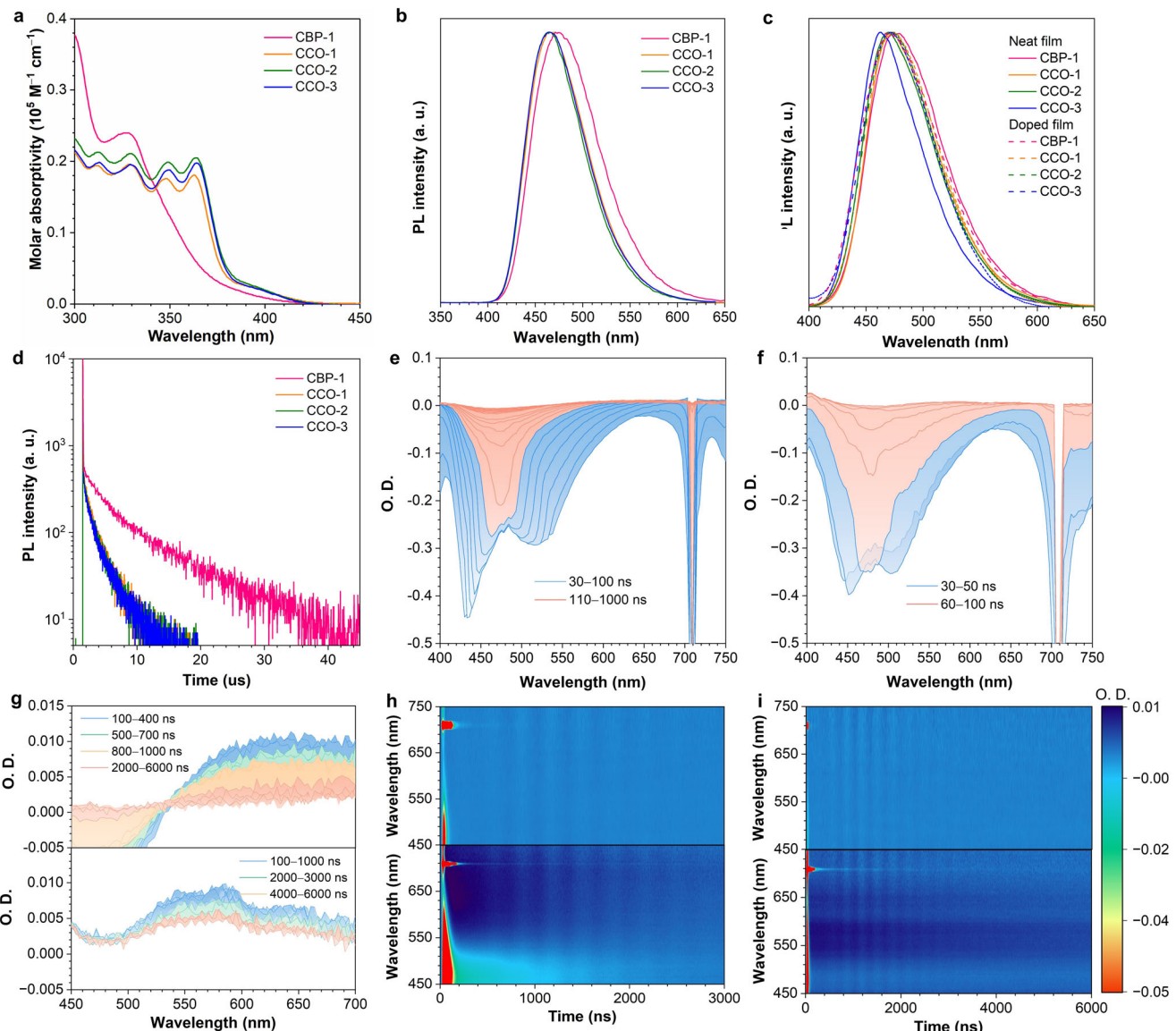

**Fig. 3 | Photophysical characterization. a** Absorption spectra and **b** PL spectra in toluene solutions ($10^{-5}$ M). **c** PL spectra in neat films (solid line) and 20 wt% doped films (dash line). **d** Temperature-dependent PL decay curves of 20 wt% doped films, measured under nitrogen. Transient absorption spectra of **e** CCO-1 and **f** CBP-1 on nanosecond timescales in oxygen-free toluene solutions ($10^{-5}$ M). Excitation wavelength: 355 nm. **g** Transient absorption spectra of CCO-1 (top line) and CBP-1 (bottom line) on microsecond timescales in oxygen-free toluene solution ($10^{-5}$ M). Transient absorption contour maps of **h** CCO-1 and **i** CBP-1 in toluene solutions under air (top line) and nitrogen (bottom line) atmosphere ($10^{-5}$ M).

the π–π* transitions, which are obviously red-shifted in comparison with that of CBP-1 due to the expansion of molecular conjugation (Fig. 3). Apparent absorption tails are observed between 380 and 430 nm stemming from transitions from SXAC donor to CCO-α acceptor. The intensities of the absorption tails of CCO-1, CCO-2 and CCO-3 are stronger than that of CBP-1, suggesting they have larger $f$ values, consistent with the theoretical calculation results. The PL spectra of CCO-1, CCO-2 and CCO-3 have peaks at about 465 nm and full width at half maximum (FWHM) of 69 nm, which are blue-shifted by 10 nm and narrowed by 13 nm, respectively, compared with those of CBP-1. In different polar solvents, such as tetrahydrofuran, dichloromethane and acetonitrile, CCO-1 shows smaller red-shifts in PL spectra than CBP-1, validating its weaker D-A interaction (Supplementary Fig. 16). The $\Phi_{PL}$s of CCO-1, CCO-2 and CCO-3 in toluene solutions are 70%, 72% and 69%, respectively, much higher than that of CBP-1 (25%), evidencing the enhanced rigidity of CCO-α has successfully reduced molecular motion and lowered nonradiative energy dissipation.

The neat films of CCO-1, CCO-2 and CCO-3 exhibit blue PL emissions with peaks at 471, 469 and 464 nm, and $\Phi_{PL}$s of 68%, 72% and 66%, respectively. The PL spectrum is blue-shifted progressively by adding *tert*-butyl groups, which is attributed to the reduced intermolecular π–π interactions. The 20 wt% doped films of these materials in diphenyl-4-triphenylsilylphenyl-phosphine oxide (DPEPO) host show stronger blue PL emissions peaking at about 470 nm and excellent $\Phi_{PL}$s reaching 99%. By contrast, CBP-1 shows slightly red-shifted PL spectra with wider FWHM by about 6 nm and lower $\Phi_{PL}$s than CCO-1 no matter in neat or doped films. The neat films of CCO-1, CCO-2 and CCO-3 have small $\Delta E_{st}$s of 63–66 meV, while their doped films hold even smaller $\Delta E_{st}$s of 33 meV. A low doping concentration is also found to be conducive to reducing $\Delta E_{st}$s of these materials. Taking CCO-1 for instance, its $\Delta E_{st}$ is decreased from 45 to 33 meV by lowering doping concentration from 40 to 20 wt% in DPEPO host (Supplementary Figs. 17–19 and Table 3). This phenomenon is actually associated with the different transition characteristics of S₁ and T₁ states, which induce

**Table 1 | Photophysical properties of CBP-1, CCO-1, CCO-2 and CCO-3**

| emitter | toluene[a] | | | | | neat film/doped film[b] | | | | | | |
|---|---|---|---|---|---|---|---|---|---|---|---|---|
| | $\lambda_{abs}$[c] (nm) | $\lambda_{em}$[d] (nm) | $\Phi_{PL}$[e] (%) | $\tau_{PF}$[f] (ns) | $\tau_{DF}$[g] (μs) | $\lambda_{em}$[d] (nm) | $\Phi_{PL}$[e] (%) | $\Delta E_{ST}$[h] (meV) | $\tau_{PF}$[f] (ns) | $\tau_{DF}$[g] (μs) | $k_r$[i] (×10$^7$ s$^{-1}$) | $k_{RISC}$[j] (×10$^5$ s$^{-1}$) |
| CBP-1 | 327 | 475 | 25 | 27 | 4.35 | 476/473 | 63/89 | 75/33 | 21/25 | 5.41/7.55 | 1.58/1.36 | 3.72/3.43 |
| CCO-1 | 363 | 466 | 70 | 25 | 1.64 | 471/470 | 68/99 | 63/33 | 22/24 | 1.69/1.91 | 2.34/1.54 | 7.68/13.81 |
| CCO-2 | 364 | 465 | 72 | 25 | 1.54 | 469/470 | 72/99 | 64/33 | 20/24 | 1.61/1.80 | 2.61/1.66 | 8.12/13.74 |
| CCO-3 | 364 | 465 | 69 | 23 | 1.54 | 464/469 | 66/99 | 66/34 | 19/23 | 1.63/1.91 | 2.73/1.58 | 7.73/14.16 |

[a]Measured in toluene solutions (10$^{-5}$ M) bubbled with nitrogen. [b]Doped films with the concentration of 20 wt%. [c]Absorption maximum. [d]PL peak wavelength. [e]Absolute photoluminescence quantum yield. [f]Lifetime of prompt fluorescence. [g]Lifetime of delayed fluorescence. [h]Energy gap between S$_1$ and T$_1$ states. [i]Radiative decay rate. [j]Reverse intersystem crossing rate.

different variations of S$_1$ and T$_1$ energy levels in response to polarity change of microenvironments at different doping concentrations[15,35,36] (Supplementary Table 4).

In neat films and 20 wt% doped films, CCO-1, CCO-2 and CCO-3 exhibit prominent delayed fluorescence with short lifetimes ($\tau_{DF}$s) of 1.61 − 1.69 μs and 1.80 − 1.91 μs, respectively, close to those in toluene solutions (1.54–1.64 μs) but much shorter than those of CBP-1 (5.41 and 7.55 μs) (Table 1, Fig. 3d and Supplementary Figs. 20–22, Table 5). The $k_{RISC}$s of CCO-1, CCO-2 and CCO-3 in 20 wt% doped films are in the range of 13.74 − 14.16 × 10$^5$ s$^{-1}$, being more than 5-fold larger than that of CBP-1 (3.43 × 10$^5$ s$^{-1}$). Concerning the same $\Delta E_{ST}$s (33 meV) of CCO-1 and CBP-1, the double channel RISC process opened by vibronic coupling between T$_1$ and T$_2$ states should account for the larger $k_{RISC}$ of CCO-1. They show decreased $k_{RISC}$s of 11.6 − 11.9 × 10$^5$ s$^{-1}$ in 40 wt% doped films and even smaller $k_{RISC}$s of 7.68 − 8.12 × 10$^5$ s$^{-1}$ in neat films (Supplementary Table 6). This tendency can be ascribed to the increased $\Delta E_{ST}$s in highly doped films and neat films. The $k_r$s of CCO-1, CCO-2 and CCO-3 gradually increase from 1.54 − 1.66 × 10$^7$ s$^{-1}$ in 20 wt% doped films, to 1.86 − 1.87 × 10$^7$ s$^{-1}$ in 40 wt% doped films, and to 2.34 − 2.73 × 10$^7$ s$^{-1}$ in neat films. Besides, benefitting from the larger $f$ values, CCO-1, CCO-2 and CCO-3 also have larger $k_r$s than CBP-1, which help to suppress SSA process.

To gain in-depth insights into the decay process of the excited states, transient absorption (TA) spectra of these materials are measured in toluene solutions. Under 355 nm pulsed laser excitation, a stimulated emission (SE) signal appears in the range of 400 − 650 nm. In comparison with the TA spectra measured before bubbling nitrogen, a new emission peak is observed at 515 nm at 30 ns after bubbling nitrogen, by which the quenching of triplet states by oxygen is alleviated (Fig. 3e, f and Supplementary Figs. 23, 24). Besides, from 400 to 600 nm, there are no obvious emission peaks of CCO-$\alpha$, CBP-$\alpha$ and SXAC, indicating that the SE signals do not stem from the LE emission (Supplementary Figs. 25, 26). Therefore, the peak at 515 nm is assigned to delayed fluorescence, and the peak at about 440 nm belongs to prompt fluorescence. The temporal dynamics feature of prompt and delayed fluorescence peaks is presented in Supplementary Information. The delayed fluorescence of CCO-1 maintains a high intensity at 30 − 100 ns, on account of the continuous up-conversion of triplet states to singlet states, while the delayed fluorescence intensity of CBP-1 decreases sharply, suggesting there are not enough triplet states to replenish the singlet states to undergo rapid deexcitation. Furthermore, CCO-1 and CBP-1 have long-lived excited state absorption (ESA) signals at 500 − 750 nm after bubbling nitrogen (Fig. 3h, i). The lifetimes of this state are fitted to be 1.59 and 5.53 μs for CCO-1 and CBP-1, respectively, at the maximum absorption peaks, which are close to their $\tau_{DF}$s measured by the transient PL decay spectra (1.64 and 4.35 μs for CCO-1 and CBP-1, respectively) (Supplementary Fig. 27). So, these ESA signals can be assigned to the absorption of triplet states participated in RISC process. The absorption signal intensity of CCO-1 declines apparently at 500 ns, indicating a large number of triplet states have been converted to singlet states in this time scale, while considerable absorption signal intensity of CBP-1 remains until

2000 ns (Fig. 3g). These findings further validate that CCO-1 has a faster RISC process than CBP-1.

**Electrochemical behavior and carrier transport property**
CCO-1, CCO-2 and CCO-3 hold good electrochemical stability, as evidenced by the reversible oxidation and reduction processes in cyclic voltammetry experiment. From the onsets of oxidation and reduction waves, their energy levels of HOMOs and LUMOs are calculated as about –5.43 and –2.71 eV, respectively, which are slightly decreased relative to those of CBP-1 (HOMO = –5.39 eV; LUMO = –2.66 eV), and their HOMO and LUMO energy gaps are similar (-2.72 eV). Moreover, their carrier transport behaviors are assessed by the space-charge limited current (SCLC) method (Supplementary Fig. 28). Hole- and electron-only devices are fabricated with the configurations of ITO/ TAPC (50 nm)/emitter (20 nm)/TAPC (40 nm)/Al and ITO/TmPyPB (50 nm)/emitters (20 nm)/TmPyPB (40 nm)/LiF (1 nm)/Al, respectively, where 1,10-bis(di-4-tolylaminophenyl)cyclohexane (TAPC), and 1,3,5-tri(m-pyrid-3-yl-phenyl)benzene (TmPyPB) are adopted as buffer layers to shield off electrons and holes, respectively. In comparison with CBP-1, CCO-1 has improved balance for hole and electron transports, and its hole and electron mobilities are kept almost equal at varied voltages. Besides, from CCO-1 to CCO-2 and to CCO-3, the electron mobility is reduced slightly, which is attributed to the increment of the interplane distance of face-to-face stacked CCO-$\alpha$ moieties that dominate electron transport[37]. The hole mobilities of CCO-2 and CCO-3 are greatly decreased relative to that of CCO-1, probably due to different stacking patterns of SXAC groups that govern hole transport (Fig. 1 and Supplementary Fig. 5).

**Electroluminescence performance**
To evaluate the EL performance of CCO-1, CCO-2 and CCO-3, OLEDs with the configuration of ITO/HATCN (5 nm)/TAPC (50 nm)/TCTA (5 nm)/$m$CP (5 nm)/EML (20 nm)/DPEPO (5 nm)/TmPyPB (30 nm)/LiF (1 nm)/Al are fabricated (Fig. 4a), in which the emitting layer (EML) is composed of their neat films or doped films with DPEPO host (doping concentration = 10, 15, 20, 30, 40 and 50 wt%). The functional layers hexaazatriphenylenehexacabonitrile (HATCN), TAPC, tris[4-(carbazol-9-yl)phenyl]amine (TCTA) and TmPyPB work as hole injection, hole transport, hole buffer and electron transport layers, respectively. Given the high T$_1$ energy levels of $m$CP (2.9 eV) and DPEPO (3.0 eV), they are adopted as exciton-blocking layers to confine excitons within EML. The devices of CBP-1 with the same configurations are also prepared for comparison. The nondoped devices of CCO-1, CCO-2 and CCO-3 are turned on at low voltages of 2.8–2.9 eV, and exhibit strong blue lights with EL peaks at 470 nm (CIE$_{x,y}$ = 0.14, 0.20), 464 nm (CIE$_{x,y}$ = 0.14, 0.17) and 456 nm (CIE$_{x,y}$ = 0.14, 0.13), respectively. These EL emissions are slightly blue-shifted along with the introduction of $tert$-butyl group(s), and become much bluer than that of CBP-1 (474 nm; CIE$_{x,y}$ = 0.17, 0.25) (Supplementary Table 8). Thanks to the large $k_r$ and $k_{RISC}$ and balanced carrier transport, the maximum $\eta_{ext}$ of CCO-1 reaches 21.6%, and the $\eta_{ext}$ at 1000 cd m$^{-2}$ still remains at 20.6%, corresponding to a very small efficiency roll-off of 4.5%. Apparently, the nondoped device of CCO-1 is among the best-nondoped OLEDs

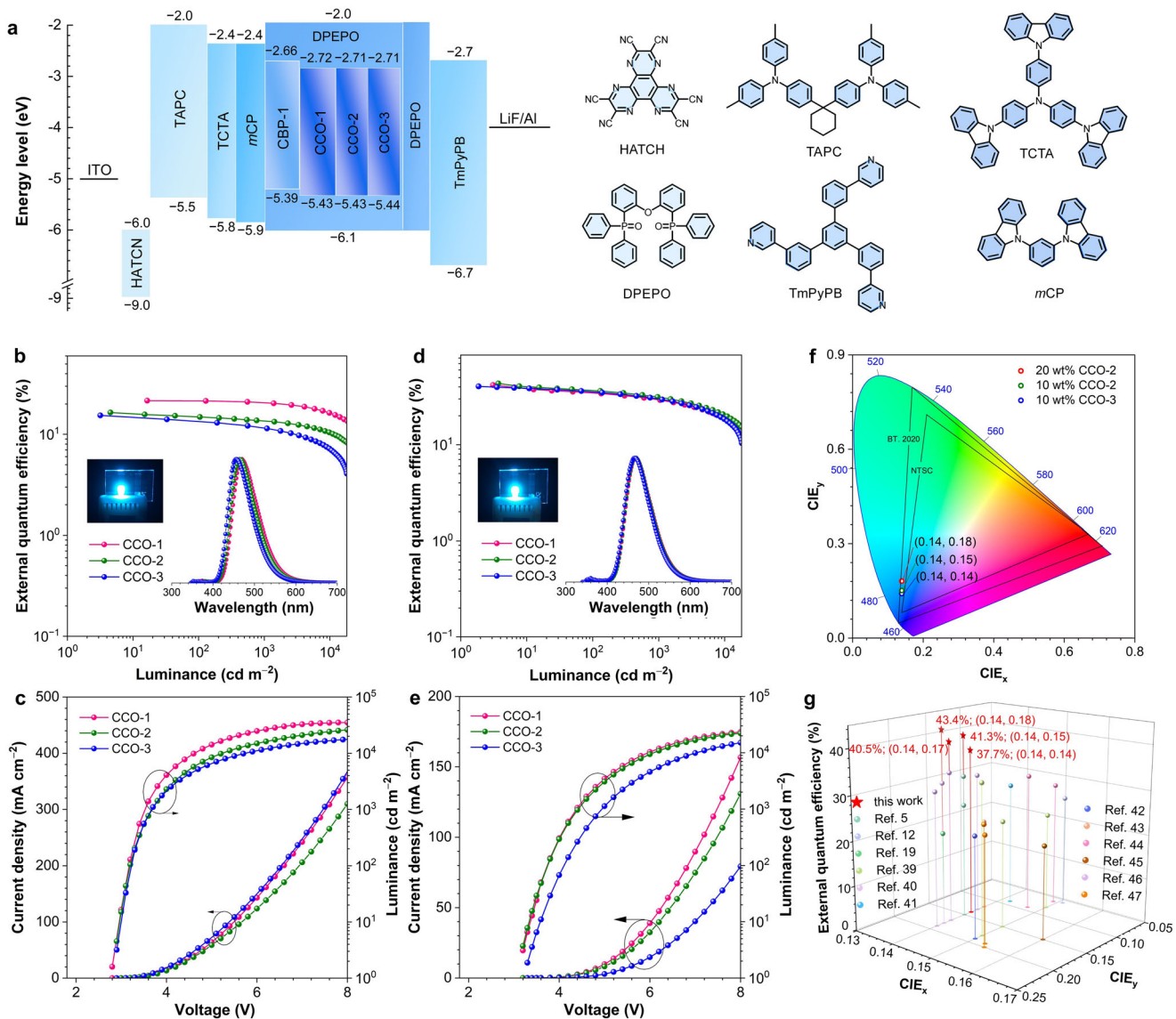

**Fig. 4 | EL performance of nondoped and doped OLEDs. a** Device architecture, energy diagram and functional layers of the OLEDs based on the new materials. Plots of **b, d** external quantum efficiency–luminance and **c, e** luminance–voltage–current density of the nondoped and doped OLEDs. Insets in planes **b, d**: EL spectra and photos (CCO-2) of the nondoped and doped OLEDs at 5 V. **f** CIE color coordinates of the doped devices based on 20 wt% CCO-2, 10 wt% CCO-2 and 10 wt% CCO-3. **g** External quantum efficiency versus CIE coordinates of representative blue and deep-blue OLEDs.

with blue and deep-blue EL emissions (CIE$_y$ ≤ 0.20) reported so far[12,38]. The maximum $\eta_{ext}$ of CBP-1 is 20.5%, but the efficiency roll-off at 1000 cd m$^{-2}$ is increased to 19.4%, due to its relatively smaller $k_r$ and $k_{RISC}$. CCO-2 and CCO-3 have similar $\Phi_{PL}$, $k_r$ and $k_{RISC}$ to CCO-1, but they show lower $\eta_{ext}$s and larger efficiency roll-offs than CCO-1 (Fig. 4 and Table 2), maybe caused by their relatively unbalanced carrier transport.

With the outstanding $\Phi_{PL}$s and $k_{RISC}$s in doped films, CCO-1, CCO-2 and CCO-3 show superb EL performances in doped devices. At 20 wt% doping concentration, the EL peaks of CCO-1, CCO-2 and CCO-3 are located at 470 nm (CIE$_{x,y}$ = 0.14, 0.19), 470 nm (CIE$_{x,y}$ = 0.14, 0.18) and 466 nm (CIE$_{x,y}$ = 0.14, 0.17), respectively. The maximum $\eta_{ext}$, current efficiency ($\eta_C$) and power efficiency ($\eta_P$) are attained as high as 41.8%, 59.8 cd A$^{-1}$, 58.7 lm W$^{-1}$ for CCO-1, 43.4%, 60.7 cd A$^{-1}$ and 59.6 lm W$^{-1}$ for CCO-2, and 40.5%, 53.9 cd A$^{-1}$ and 51.3 lm W$^{-1}$ for CCO-3, respectively, which are much better than those of CBP-1 (33.8%, 54.0 cd A$^{-1}$ and 53.0 lm W$^{-1}$) (Supplementary Table 8). The $\eta_{ext}$s of CCO-1, CCO-2 and CCO-3 at 1000 cd m$^{-2}$ luminance are still maintained as high as of 29.4%, 31.3% and 30.0%, respectively, indicative of good efficiency

stability. To the best of our knowledge, the eximious $\eta_{ext}$s of 40.5%–43.4% are the highest values for blue and deep-blue OLEDs with CIE$_y$ ≤ 0.20 reported so far, eloquently demonstrating the great success of our molecular design (Supplementary Table 7)[5,12,19,39–47].

The horizontal transition dipole orientation of these materials in 20 wt% doped films are investigated by using the angle-dependent $p$-polarized PL spectra (Supplementary Fig. 30). It is found that CBP-1 has a good $\Theta_{//}$ of 81.2%, owing to the presence of spiro SXAC moiety, while CCO-1, CCO-2 and CCO-3 show much superior $\Theta_{//}$s of 91.7%, 93.6% and 90.1%, respectively, validating that the enhancement in molecular planarity contributes significantly to promote horizontal transition dipole orientation of these materials. With the data of $\Theta_{//}$, reflex index and active layer thickness, the $\eta_{out}$s of CCO-1, CCO-2 and CCO-3 are calculated as high as 43.0%, 43.9% and 42.3%, respectively, approaching the theoretical limit of $\eta_{out}$ (~45%)[48], and higher than that of CBP-1 (38.0%). Assuming that exciton utilization and exciton recombination efficiency are 100%, the corresponding theoretical maximum $\eta_{ext}$s of CCO-1, CCO-2 and CCO-3 are 42.6%, 43.5% and 41.9%, respectively, in good agreement with device data. Clearly,

**Table 2 | EL performances of the new molecules in nondoped, doped and HF OLEDs**

| emitter | $V_{on}$ (V)[a] | $\eta_{C,max}$ (cd A$^{-1}$)[b] | $\eta_{P,max}$ (lm W$^{-1}$)[c] | $\eta_{ext,max}$ (%)[d] | $\eta_{ext,1000}$ (%)[e] | Roll-off (%)[f] | $L_{max}$ (cd m$^{-2}$)[g] | CIE$_{x,y}$[h] | $\lambda_{EL}$ (nm)[i] |
|---|---|---|---|---|---|---|---|---|---|
| nondoped device | | | | | | | | | |
| CCO-1 | 2.8 | 36.6 | 41.1 | 21.6 | 20.6 | 4.5 | 35190 | (0.14, 0.20) | 470 |
| CCO-2 | 2.9 | 22.0 | 23.8 | 16.5 | 13.6 | 17.4 | 28880 | (0.14, 0.17) | 464 |
| CCO-3 | 2.9 | 16.9 | 18.3 | 15.4 | 11.6 | 24.8 | 18840 | (0.14, 0.13) | 456 |
| doped device | | | | | | | | | |
| 10 wt% CCO-1 | 3.6 | 49.3 | 43.0 | 37.8 | 22.7 | 39.8 | 9111 | (0.14, 0.16) | 464 |
| 15 wt% CCO-1 | 3.5 | 50.7 | 45.5 | 37.8 | 25.0 | 33.9 | 11590 | (0.14, 0.17) | 468 |
| 20 wt% CCO-1 | 3.2 | 59.8 | 58.7 | 41.8 | 29.4 | 29.6 | 24560 | (0.14, 0.18) | 470 |
| 30 wt% CCO-1 | 3.0 | 63.8 | 66.8 | 40.9 | 34.1 | 16.6 | 50690 | (0.14, 0.21) | 472 |
| 40 wt% CCO-1 | 2.9 | 63.8 | 69.1 | 39.8 | 33.5 | 15.8 | 54270 | (0.14, 0.22) | 474 |
| 50 wt% CCO-1 | 2.8 | 63.8 | 71.5 | 39.5 | 33.4 | 15.4 | 56510 | (0.14, 0.22) | 474 |
| 10 wt% CCO-2 | 3.7 | 50.4 | 42.8 | 41.3 | 22.4 | 45.7 | 7396 | (0.14, 0.15) | 462 |
| 15 wt% CCO-2 | 3.5 | 53.8 | 48.3 | 41.4 | 27.4 | 34.0 | 12570 | (0.14, 0.16) | 464 |
| 20 wt% CCO-2 | 3.2 | 60.7 | 59.6 | 43.4 | 31.3 | 27.8 | 24400 | (0.14, 0.18) | 470 |
| 30 wt% CCO-2 | 3.0 | 58.4 | 61.2 | 41.0 | 32.1 | 21.8 | 31613 | (0.14, 0.18) | 470 |
| 40 wt% CCO-2 | 3.0 | 60.1 | 62.9 | 41.6 | 33.3 | 19.9 | 39564 | (0.14, 0.19) | 470 |
| 50 wt% CCO-2 | 2.9 | 61.8 | 67.0 | 41.6 | 33.7 | 19.0 | 47777 | (0.14, 0.19) | 470 |
| 10 wt% CCO-3 | 3.8 | 44.4 | 36.7 | 37.7 | 20.2 | 46.4 | 5833 | (0.14, 0.14) | 462 |
| 15 wt% CCO-3 | 3.5 | 50.8 | 45.5 | 39.8 | 26.8 | 32.6 | 11570 | (0.14, 0.16) | 464 |
| 20 wt% CCO-3 | 3.3 | 53.9 | 51.3 | 40.5 | 30.0 | 25.8 | 18510 | (0.14, 0.17) | 466 |
| 30 wt% CCO-3 | 3.1 | 59.1 | 59.9 | 41.4 | 33.1 | 20.1 | 36700 | (0.14, 0.18) | 470 |
| 40 wt% CCO-3 | 3.0 | 58.9 | 61.7 | 40.3 | 33.6 | 16.6 | 44520 | (0.14, 0.19) | 470 |
| 50 wt% CCO-3 | 2.9 | 58.8 | 63.6 | 39.9 | 33.4 | 16.1 | 49260 | (0.14, 0.19) | 470 |
| HF device | | | | | | | | | |
| DABNA-NP-TB | 3.1 | 30.0 | 30.4 | 32.5 | 20.8 | 35.8 | 21670 | (0.14,0.10) | 456 |
| BN2 | 3.6 | 158.8 | 136.8 | 37.6 | 13.3 | 64.7 | 33930 | (0.32,0.64) | 540 |

[a]Turn-on voltage at 1 cd m$^{-2}$. [b]Maximum current efficiency. [c]Maximum power efficiency. [d]Maximum external quantum efficiency. [e]External quantum efficiency at 1000 cd m$^{-2}$. [f]External quantum efficiency roll-off at 1000 cd m$^{-2}$. [g]Maximum luminance. [h]Commission Internationale de l'Eclairage coordinates. [i]EL peak wavelength.

these excellent $\eta_{out}$s have significantly contributed to boost the EL efficiencies of the devices.

In addition, CCO-1, CCO-2 and CCO-3 can also provide strong blue and deep-blue EL emissions at varied doping concentrations (Table 2). For example, CCO-2 at 10 wt% doping concentration radiates strong deep-blue light peaking at 462 nm (CIE$_{x,y}$ = 0.14, 0.15) and attains the maximum $\eta_{ext}$ of 41.3%. At 50 wt% doping concentration, the EL peak moves to 470 nm (CIE$_{x,y}$ = 0.14, 0.19) and the maximum $\eta_{ext}$ remains as 41.6%. Notably, by increasing doping concentration from 10 to 50 wt%, the efficiency roll-offs decline gradually from 39.8% to 15.4% for CCO-1, 45.7% to 19.0% for CCO-2, and 46.4% to 16.1% for CCO-3. At high doping concentrations, $k_{RISC}$ is decreased, but $k_r$ is increased, which can effectively alleviate SSA (Supplementary Table 5). Besides, taking CCO-2 for instance, the carrier mobility of 40 wt% doped film is more unbalanced than that of 20 wt% one, but the 40 wt% doped device has a much smaller efficiency roll-off. This result indicates that TPA should not be the dominant factor for the exciton loss (Supplementary Fig. 31). Thus, the large efficiency roll-offs at low doping concentrations suggest that SSA is a main factor undermining efficiency stability when $k_{RISC}$ is large enough.

Currently, HF OLEDs that utilize D-A type TADF materials as sensitizers for multi-resonance TADF (MR-TADF) dopants with narrow FWHM emissions are considered as a promising technique to fulfill the requirements for ultra-high-definition display because they can achieve high EL efficiency and high color purity simultaneously. Based on Förster energy transfer from sensitizer to dopant, HF OLEDs can make full use of the advantages of high exciton utilization of the sensitizers and the narrow emissions of the dopants. In view of the outstanding EL performance of CCO-2, it is chosen as an example for the fabrication of deep-blue and green HF OLEDs with the configuration of

ITO/HATCN (5 nm)/TAPC (50 nm)/TCTA (5 nm)/mCP (5 nm)/EML (20 nm)/DPEPO (5 nm)/TmPyPB (30 nm)/LiF (1 nm)/Al (Fig. 5). In EML, the commercially available deep-blue and green MR-TADF materials of DABNA-NP-TB[49,50] and BN2[51] are adopted as dopants, whose absorption spectra are partially overlapped with the PL spectrum of CCO-2 (Supplementary Fig. 32). And CCO-2 doped in 9-(3-(9H-carbazol-9-yl) phenyl)-9H-3,9′-bicarbazole (mCPBC) and 2,6-bis(4-(9H-carbazol-9-yl) phenyl)pyridine (2,6-DCzPPy) hosts with a doping concentration of 40 wt% functions as the sensitizer for DABNA-NP-TB and BN2 (doping concentration = 1 wt%), respectively. The HF OLED based on DABNA-NP-TB radiates high-purity deep-blue light with an EL peak at 456 nm (CIE$_{x,y}$ = 0.14, 0.10; FWHM = 31 nm) and a maximum luminance reaching up to 21670 cd m$^{-2}$. Meanwhile, a remarkable $\eta_{ext}$ of 32.5% is attained, verifying this device is one of the best HF OLEDs with EL peaks < 460 nm[47,52,53]. The HF OLED based on BN2 shows strong green light with an EL peak at 540 nm (CIE$_{x,y}$ = 0.32, 0.64; FWHM = 41 nm) and a maximum luminance as high as 33930 cd m$^{-2}$. Impressive $\eta_C$, $\eta_P$ and $\eta_{ext}$ of 158.8 cd A$^{-1}$, 136.8 lm W$^{-1}$ and 37.6% are achieved, being the highest EL efficiencies for green MR-TADF materials[54,55].

The operational lifetimes of these HF OLEDs are also investigated. Two kinds of device structures are optimized for DABNA-NP-TB (HF1 and HF2, device structures are listed in Supplementary Materials). Devices HF1 and HF2 display deep blue lights with CIE$_{x,y}$ of (0.14, 0.12) and (0.14, 0.10) and provide maximum $\eta_{ext}$s of 23.5% and 17.0%, respectively (Supplementary Fig. 33 and Table 9). The operational lifetimes ($LT_{50}$) for the luminance to decay to 50% of the initial luminance of 1000 cd m$^{-2}$ are 9.8 and 21.9 h for HF1 and HF2, respectively, and the estimated $LT_{50}$ values at 100 cd m$^{-2}$ initial luminance are calculated as 621 and 1099 h, respectively, with a decay factor of 1.7. It is worth noting that, at 100 cd m$^{-2}$ initial luminance, the $LT_{80}$ values of

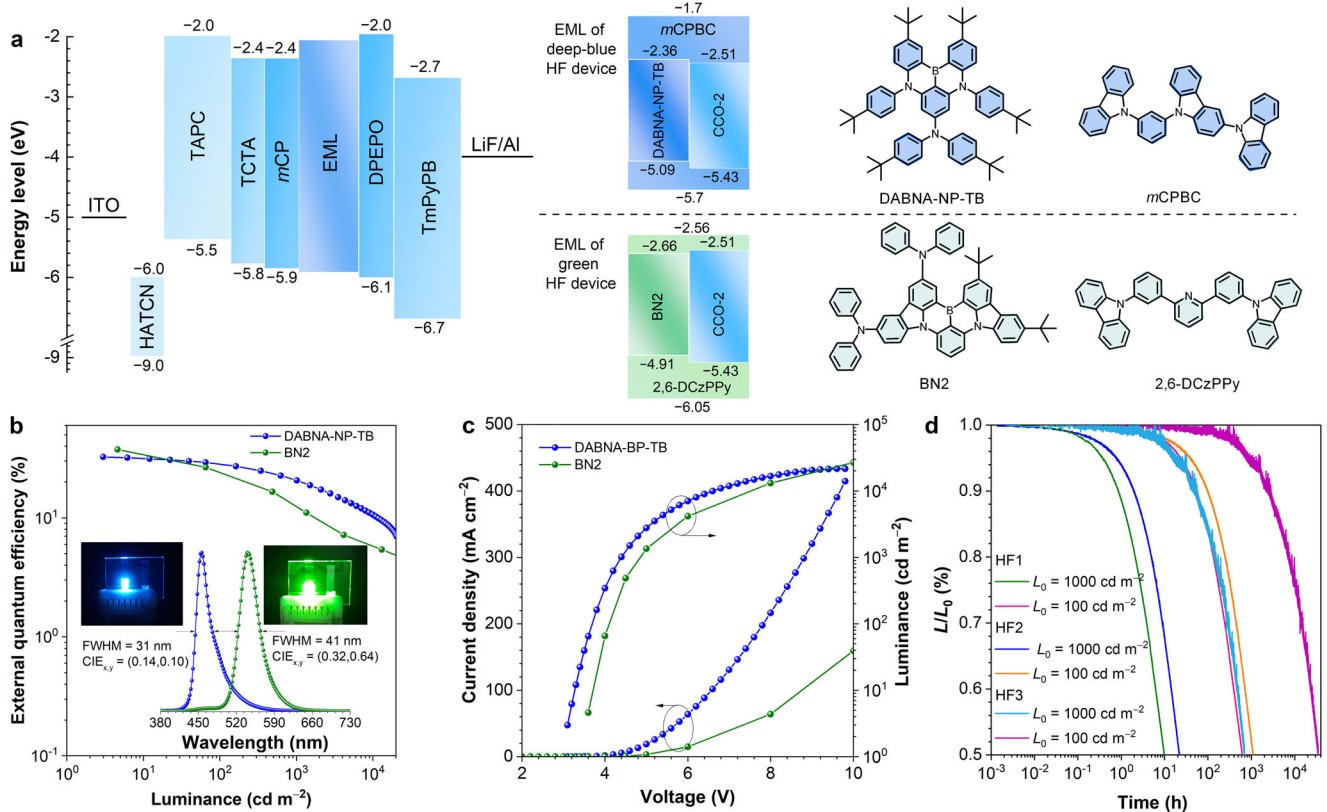

**Fig. 5 | EL performance of HF OLEDs. a** Device architecture, energy diagram and molecular structures of deep-blue and green HF OLEDs. The LUMO energy levels of DABNA-NP-TB and BN2 are the reported data, calculated by the HOMO energy level plus optical band gap[49,51]. To maintain consistency of the calculation method of energy levels, the LUMO energy level of CCO-2 is recalculated as −2.51 eV by the reported method. Plots of **b** external quantum efficiency–luminance and **c** current density–voltage–luminance of HF OLEDs. Inset in planes **b**: EL spectra and photos of HF OLEDs at 5 V. **d** Operational lifetimes of devices HF1, HF2 and HF3.

HF1 and HF2 are 133 and 250 h, respectively, significantly advanced relative to the reported data ($\eta_{ext}$ = 19.5%; $LT_{80}$ = 19 h) for DABNA-NP-TB[50]. Device HF3, optimized for BN2, shows green light (CIE$_{x,y}$ = 0.33, 0.63) with a maximum $\eta_{ext}$ of 29.7%, and the $LT_{50}$ at 1000 cd m$^{-2}$ initial luminance is 692 h, from which the $LT_{50}$ at 100 cd m$^{-2}$ initial luminance is obtained as 34669 h. These operational lifetimes acquired in common laboratory are virtually good (Supplementary Table 10), and could be further improved under better conditions with robust functional materials in industry.

## Discussion

In summary, a weak electron acceptor CCO-$\alpha$ with a rigid polycyclic structure is generated by ring-closure in carbazole-benzoyl framework with an oxygen linkage. And the combination of CCO-$\alpha$ with electron donor SXAC furnish a series of robust blue and deep-blue delayed fluorescence materials (CCO-1, CCO-2 and CCO-3). The ring-closed structure by oxygen atom bridge has no contribution to SOC, but results in higher rigidity and planarity, better conjugation and lower electron-withdrawing ability and can influence excited-state energy levels of the molecules, leading to reduced $\Delta E_{st}$s. The introduction of *tert*-butyl group(s) can suppress tight molecular packing and weaken intermolecular interaction, beneficial for acquiring blue emissions in solid state. In comparison with the control material CBP-1 with an unclosed carbazole-benzoyl acceptor, CCO-1, CCO-2 and CCO-3 own improved structural rigidity and weakened D-A interactions, and exhibit faster radiative transition and RISC process with larger $k_r$s and $k_{RISC}$s. In consequence, they emit stronger and bluer delayed fluorescence with shorter $\tau_{DF}$s and higher $\Phi_{PL}$s reaching 99%. Moreover, the improved molecular planarity of CCO-$\alpha$ promotes horizontal transition dipole orientation of the materials, leading to higher $\Theta_{//}$s of 93.6%

and thus remarkable $\eta_{out}$s of 43.9%. Owing to these distinguished merits, they function outstandingly as emitters in OLEDs, providing blue (CIE$_{x,y}$ = 0.14, 0.18) and deep-blue (CIE$_{x,y}$ = 0.14, 0.15) lights with record-beating $\eta_{ext}$s of 43.4% and 41.3%, respectively. And small efficiency roll-offs are achieved by suppressing SSA and TTA with accelerated radiative transition and RISC process. Beside, by utilizing CCO-2 as sensitizer for MR-TADF dopants, high-performance deep-blue (CIE$_{x,y}$ = 0.14, 0.10) and green (CIE$_{x,y}$ = 0.32, 0.64) HF OLEDs are realized with state-of-the-art $\eta_{ext}$s of 32.5% and 37.6%, respectively. The stability of these HF OLEDs is also improved, with the $LT_{50}$ at 100 cd m$^{-2}$ initial luminance of 1099 h for deep-blue HF device and 34669 h for green one. Thus, we believe that these tailor-made blue and deep-blue delayed fluorescence materials are promising candidats as emitters and sensitizers for the fabrication of high-performance OLEDs, and the presented design strategy and structure-property insights are beneficial for the exploration of luminescent materials for display applications.

## Methods

### Materials and instruments

All the chemicals and reagents were purchased from commercial sources and used as received without further purification. The final products were subjected to vacuum sublimation to further improve purity before PL and EL properties investigations. $^1$H and $^{13}$C NMR spectra were measured on Bruker AV 500 spectrometer or Bruker AV 400 spectrometer in CDCl$_3$, CD$_2$Cl$_2$ or C$_2$D$_6$SO at room temperature. High-resolution mass spectra (HRMS) were recorded on a GCT premier CAB048 mass spectrometer operating in MALDI-TOF mode. Thermogravimetric analysis (TGA) was carried on a TA TGA Q5000 under dry nitrogen at a heating rate of 20 °C min$^{-1}$. Thermal transition was

investigated by a TA DSC Q1000 under dry nitrogen at a heating rate of $10\,°C\,min^{-1}$. Single crystals of CBP-1, CCO-1, CCO-2 and CCO-3 are cultured in dichloromethane/*n*-hexane, chloroform/*n*-hexane, deuterated chloroform and chloroform/ethanol, respectively (Supplementary Data 1–4). Single crystal X-ray diffraction intensity data were collected at 173 K on a Bruker−Nonices Smart Apex CCD diffractometer with graphite monochromated MoKα radiation. Processing of the intensity data was carried out using the SAINT and SADABS routines, and the structure and refinement were conducted using the SHELTL suite of X-ray programs (version 6.10). UV-vis absorption spectra were measured on a Shimadzu UV-2600 spectrophotometer ($f \approx 4.33 \times 10^{-9}\, \varepsilon\, \Delta v_{1/2}$; $\varepsilon$ is absorption intensity; $\Delta v_{1/2}$ is wavenumber at half absorption strength). PL spectra were recorded on a Horiba Fluoromax-4 spectrofluorometer. PL quantum yields were measured using a Hamamatsu absolute PL quantum yield spectrometer C11347 Quantaurus_QY. Transient PL decay spectra were measured using Quantaurus-Tau fluorescence lifetime measurement system (C11367-03, Hamamatsu Photonics Co., Japan). The transient absorption spectra were recorded by a commercial TA system (Transpec-NS, CIS) equipped with a nanosecond pulse laser (Opolette 355 LD, 7 ns, 20 Hz) as the pump source and a broadband laser-driven light source (EQ99X LDLS) as the probe source. The detector output was fed to a digital oscilloscope (PicoScope 5000), which acquired the waveform and stored it for eventual data processing. The pump wavelength was 355 nm, and the measurement was carried out at room temperature. The angular-dependent spectra were measured with an ARM angle-resolved spectroscopy system (IdeaOptics Instruments). Cyclic voltammograms were measured in a solution of tetra-*n*-butylammonium hexafluorophosphate ($n$Bu$_4$NPF$_6$, 0.1 M) in dichloromethane (anodic) and dimethylformamide (cathodic) at a scan rate of $50\,mV\,s^{-1}$, using a platinum wire as the auxiliary electrode, a glass carbon disk as the working electrode and Ag/Ag$^+$ as the reference electrode, standardized for the redox couple ferricenium/ferrocene (Fc/Fc$^+$) (HOMO = $-[E_{onset}{}^{ox} + 4.8]$ eV, and LUMO = $-[E_{onset}{}^{re} + 4.8]$ eV). $E_{onset}{}^{ox}$ and $E_{onset}{}^{re}$ represent the onset oxidation and reduction potentials relative to Fc/Fc$^+$, respectively.

### Device fabrication and characterization

The glass substrates precoated with a 90-nm layer of indium tin oxide (ITO) with a sheet resistance of 15-20 Ω per square were successively cleaned in ultrasonic bath of acetone, isopropanol, detergent and deionized water, respectively, taking 10 min for each step. Then, the substrates were completely dried in a 70 °C oven. Before the fabrication processes, in order to improve the hole injection ability of ITO, the substrates were treated by O$_2$ plasma for 10 min. The vacuum-deposited OLEDs were fabricated under a pressure of $< 5 \times 10^{-4}$ Pa in the Fangsheng OMV-FS450 vacuum deposition system. Organic materials, LiF and Al were deposited at rates of 1-2, 0.1 and $5\,A\,s^{-1}$, respectively. The effective emitting area of the device was 9 mm$^2$, determined by the overlap between anode and cathode. The luminance–voltage–current density and external quantum efficiency were characterized with a dual-channel Keithley 2614B source meter and a PIN-25D silicon photodiode except the green HF device, which were characterized with a Keithley 2400 Source Meter and a PhotoResearch PR670 spectroradiometer. The EL spectra were obtained via an Ocean Optics USB 2000+ spectrometer, along with a Keithley 2614B Source Meter except the green HF device, which were characterized with a Keithley 2400 Source Meter and a PhotoResearch PR670 spectroradiometer. All the characterizations were conducted at room temperature in ambient conditions without any encapsulation except for device lifetime measurement, as soon as the devices were fabricated. The operational lifetimes of the devices are fitted with formula $L/L_0 = \exp(-(t/\tau)^\beta)$. The curves of $L_0$ *vs.* $LT_{50}$ are fitted from $L_0{}^n LT_{50} = $ constant.

### Theoretical calculation

The ground-state geometries were optimized using the density function theory (DFT) method with PBE0 functional at the basis set level of 6-31 G (d, p), and the excited-state geometries were optimized using the time-dependent density function theory (TDDFT) method with PBE0 functional at the basis set level of 6-31 G (d, p). The $\Delta E_{ST}$s were calculated by the adiabatic excitation energy. The above calculations were performed using Gaussian16 package. The combined quantum mechanics and molecular mechanics (QM/MM) method with two-layer ONIOM approach was used to simulate the properties in the solid state. The charge of atoms in UFF force field is calculated with QEQ method, and using MM charges from the real system in the QM calculations on the model system. The NTO, MPI and ADCH were analyzed by Multiwfn[56]. Based on the obtained data from Gaussian16 package, the frequency analysis for reorganization energies ($\lambda$), Huang-Rhys factor ($S$) and theoretical $k_{RISC}$ were calculated by the MOMAP package, revision 2020A[57,58]. Geometry comparisons and RMSD values between S$_0$ and S$_1$ structures were performed by VMD 1.9.3.

### Carrier mobility measurement

The SCLC method can be described via the Mott-Gurney Eq. (3), and the carrier mobility ($\mu$) of organic semiconductors can be calculated according to the Poole−Frenkel Eq. (4), where the $\varepsilon_0$ is the free-space permittivity ($8.85 \times 10^{-14}$ C V$^{-1}$ cm$^{-1}$), $\varepsilon_r$ is the relative dielectric constant (assumed to be 3.0 for organic semiconductors), $E$ is the electric field, $\mu_0$ is the zero-field mobility, and $\gamma$ is the Poole-Frenkel factor and $L$ is the thickness of the neat film of each molecule.

$$J = \frac{9}{8}\varepsilon_0\varepsilon_r\mu\frac{E^2}{L} = \frac{9}{8}\varepsilon_0\varepsilon_r\frac{V^2}{L^3}\mu_0 e^{0.891\gamma\sqrt{\frac{V}{L}}} \tag{3}$$

$$\mu = \mu_0 e^{\gamma\sqrt{E}} \tag{4}$$

By fitting the current density–voltage curves in SCLC region according to Eq. (3)[59], the $\mu_0$ and $\gamma$ values are obtained, thus generating the field-dependent carrier mobility by Eq. (4).

## Data availability

All the data generated in this study are provided in the Source Data file. Crystallographic data for the structures reported in this Article have been deposited at the Cambridge Crystallographic Data Centre, under deposition numbers CCDC 2209209 (CBP-1), 2209210 (CCO-1), 2209211 (CCO-2) and 2209436 (CCO-3). Copies of the data can be obtained free of charge via https://www.ccdc.cam.ac.uk/structures/. Source data are provided with this paper.

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

## Acknowledgements

This study is financially supported by the National Natural Science Foundation of China (21788102), the GuangDong Basic and Applied Basic Research Foundation (2023B1515040003, 2022A1515010315 and 2019B030301003) and the Fundamental Research Funds for the Central Universities. We also acknowledge HZWTECH for providing computation facilities.

## Author contributions

Z.Z. conceived the study. Y.F. synthesized and characterized the materials, measured the photophysical property and performed the theoretical simulation. H.L. fabricated and characterized the OLEDs. Y.F. and Z.Z. wrote and revised the manuscript. Z.Z. and B.Z.T. supervised the project. All authors discussed the results and commented on the manuscript.

## Competing interests

The authors declare no competing interests.
