## [Peer Review File · Nature Communications]

Realizing efficient blue and deep-blue delayed fluorescence materials with record-beating electroluminescence efficiencies of 43.4%Reviewers' Comments:

Reviewer #1:

Remarks to the Author:

See review attached

Fu et al. designed new deep blue TADF emitting materials for Organic Light-Emitting Diodes (OLEDs) applications. They carried out theoretical and spectroscopic investigations and prepared conventional as well as hyperfluorescence devices. The efficiencies of these devices are pretty high reaching more than 40% due very good molecular alignment within the device.

Overall, the device part has been well conducted but I feel that the theoretical and the photophysics parts must be revised:

Major issues

1) In the introduction, the authors mentioned triplet-triplet (TTA) and singlet-singlet (SSA) annihilations as the dominant sources of device degradation. They completely overlooked triplet-charge annihilation (TCA) which is clearly a major source of triplet loss. In the device section, they also mention that varying the doping ratio has an impact on SSA but they don't discuss why they interpret their data in terms of SSA and not TTA and TCA. A physical justification and an experimental proof would be highly welcome.

2) In page 4, the interpretation of the orbital evolution going from CBP0 to CCO-1 is misleading. They mentioned increase in conjugation to explain the decrease of the LUMO going from CBP0 to CCO-1 is not correct, adding an oxygen to planarize the acceptor does not result in an increase in conjugation. Regarding the evolution of the HOMO energy, I clearly do not know what they mean by CT effect. Evolution of the orbital energies should be discussed in terms of inductive and/or mesomeric electron donating or withdrawing effects and I bet the effect is mainly inductive in view of the symmetric stabilization of the HOMO and LUMO orbitals. Clearly, this paragraph has to be revised.

3) In page 4, "Due to weakened D-A interaction and thus lowered CT effect, the LE feature of CCO-1 is enhanced accordingly, indicating the overlap integral between transition orbitals will increase". I guess the authors mentioned this for the S1 state because this is absolutely not the case for T1, it is rather the opposite. The authors did not discuss why there is a drastic decrease in the overlap factor for the T1 state while this is very much crucial for the rest of the discussion in the paper (e.g. RISC rate). Overall, I think this paragraph is confusing and is missing some analysis.

3) In page 7, the evolution of the emission energy as a function of the %wt makes a reference to polarization effects but it remains quite speculative in my opinion. A qualitative analysis in terms of the dipole of the molecules could further rationalize this effect.

4) In page 7, about the analysis of the transient absorption, I think the analysis is quite misleading and I think that the experiments should be redone. It is quite funny to see two emission peaks at the early stage (below 100 ns) at 440 and 515 nm which do not match at all with the emission energy coming from steady state PL emission. Why would in any ways the 515 nm emission be the delayed fluorescence component? Usually, prompt and delayed fluorescence match quite well and the emission at a later stage around 470 nm matches quite well with the expected emission energy. I clearly doubt that the 515 nm is the delayed fluorescence. The interpretation of the TA is highly questionable, I believe 440 and 515 are actually the LE emissions of the donor and the acceptor (either one or the other) and not the one of the CT band. In my view, the reason is found in the excitation wavelength which is well above the CT-like absorption band edge. Clearly, a higher-lying excited state is probed at this excitation wavelength leading to a path on the manifold of excited states that could be very different compared to the one we would obtain when exciting the CT band.

5) In page 11, I don't understand why in the hyperfluorescence OLED device with DABNA-NP-TB, the emission comes from DABNA-NP-TB. In view of the electronic levels diagram, it is suggested that a

lower band gap is measured for CCO-2 which would suggest that energy transfer takes place from DABNA-NP-TB to CCO-2

Minor issues:

1) In page 2, it is a minor point I doubt we have a lot of red emitting TADF materials.

2) In page 3, “And the root of the mean of squared displacement (RMSD)” should be replaced by “And the root mean squared displacement (RMSD)”

Reviewer #2:

Remarks to the Author:

The manuscript by Zhao and coworkers reported novel chromeno[3,2-c]carbazol-8(5H)-one-based TADF materials. Their horizontal dipole ratio (>90%) and f value were reasonably improved by the bridging oxygen atom. The RISC process was carefully studied by quantum chemical calculations. Photophysical characterization based on temperature-dependent PL decay curves and transient absorption spectra provide useful insights for the readers. Moreover, as the authors emphasized, the maximum external quantum efficiency of more than 43% is the best value ever achieved for blue TADF materials, and should have a sufficient impact on readers and contribute to further developments in this field. However, there are several issues to be concerned as follows. I recommend careful revisions prior to publication.

1. Discussion based on the energy gaps are not sufficient for the RISC process. It would be better to conduct SOC calculations to clarify effect of the bridging oxygen atom.
2. There are many reports of blue and green HF devices using MR-TADF materials. EQE and lifetime should be compared with those reported for the HF devices as for the simple TADF devices (Table S7).
3. The operational lifetime of HF3 has only been measured partway through.
4. Given the measurement error and standard deviation between devices, there is no scientific basis for 4 significant digits for EQEs.

The followings are our point-by-point responses to the reviewers' comments and suggestions.

Reviewer #1

1) In the introduction, the authors mentioned triplet-triplet (TTA) and singlet-singlet (SSA) annihilations as the dominant sources of device degradation. They completely overlooked triplet-charge annihilation (TCA) which is clearly a major source of triplet loss. In the device section, they also mention that varying the doping ratio has an impact on SSA but they don't discuss why they interpret their data in terms of SSA and not TTA and TCA. A physical justification and an experimental proof would be highly welcome.

Response: We thank reviewer's good suggestion. As for triplet-charge annihilation (TCA), which is also known as triplet-polaron annihilation (TPA), it is mainly induced by the unbalanced carrier transport, and it is inevitable in such PN-junction devices. Hence, we do not deny that TPA also leads to the loss of triplet excitons, but we mainly focus on the suppression of TTA and SSA in this work. To verify the influence of TPA process on degeneration of device, we examined the hole and electron mobilities of 20 and 40 wt% doped films for CCO-2 by the space-charge limited current (SCLC) method on the hole- and electron-only devices with configurations of ITO/TAPC (50 nm)/emitter (20 nm)/TAPC (40 nm)/Al and ITO/ TmPyPB (50 nm)/emitter (20 nm)/TmPyPB (40 nm)/LiF (1 nm)/Al, respectively. As shown in **Fig. R1**, the carrier mobility of 40 wt% doped film is more unbalanced than that of 20 wt% one, but the 40 wt% doped device has a much smaller efficiency roll-off. This result indicates that TPA should not be the dominant factor for exciton loss in these devices. For the triplet-triplet (TTA) and singlet-singlet (SSA) annihilations, they are directly affected by k_{RISC} and k_{r} , respectively, which determine how long triplet and singlet excitons exist. According to the photophysical data (**Table R1**), it is obvious that the 40 wt% doped films with larger k_{r} s have smaller efficiency roll-offs despite the larger k_{RISC} s of 20 wt% doped films. Therefore, the SSA could be another main cause for the efficiency roll-off in addition to TTA.

Fig. R1. The characterizations of bipolar carrier mobility by SCLC method. Electric field-dependent carrier mobility of 20 and 40 wt% doped films for CCO-2 in single-carrier devices.

Table R1. Photophysical transition rates of doped films of the new molecules, and the corresponding efficiency roll-offs of doped devices at 1000 cd m⁻².

emitter	k_{r} ($\times 10^7 \text{ s}^{-1}$) ^a	k_{RISC} ($\times 10^5 \text{ s}^{-1}$) ^b	roll-off (%) ^c
20 wt% doped films			
CCO-1	1.54	13.81	29.6

CCO-2	1.66	13.74	27.8
CCO-3	1.58	14.16	25.8
40 wt% doped films			
CCO-1	1.86	11.7	15.8
CCO-2	1.87	11.6	19.9
CCO-3	1.87	11.9	16.6

^aRadiative decay rate. ^bReverse intersystem crossing rate. ^cExternal quantum efficiency roll-off at 1000 cd m⁻².

2) In page 4, the interpretation of the orbital evolution going from CBP-1 to CCO-1 is misleading. They mentioned increase in conjugation to explain the decrease of the LUMO going from CBP-1 to CCO-1 is not correct, adding an oxygen to planarize the acceptor does not result in an increase in conjugation. Regarding the evolution of the HOMO energy, I clearly do not know what they mean by CT effect. Evolution of the orbital energies should be discussed in terms of inductive and/or mesomeric electron donating or withdrawing effects and I bet the effect is mainly inductive in view of the symmetric stabilization of the HOMO and LUMO orbitals. Clearly, this paragraph has to be revised.

Response: We thank reviewer's good suggestion. Well, the inductive effect is a short-range force, and gets very weak at the third carbon. We compare the charge distributions of CBP-1 and CCO-1, and find that they have similar distributions in donors, which are listed in **Fig. R2** and **Table R2**. Hence, the evolution of HOMO energy level should not be ascribed to the inductive effect at least. Regarding the weakened D-A interaction mentioned in manuscript, it can be explained by perturbation molecular orbital theory. The weak D-A interaction leads to a weak electrostatic interaction, which would increase the energy gap between the frontier molecular orbital, manifesting as the increase of LUMO energy level (E_{LUMO}) and the decrease of HOMO energy level (E_{HOMO}) (**Fig. R3**). Indeed, the E_{HOMO} of CCO-1 is decreased by 0.058 eV relative to that of CBP-1, while the E_{LUMO} of CCO-1 is also reduced by 0.066 eV instead of being increased owing to improved conjugation of CCO- α segment, which is induced by the enhanced molecular planarity and the formation of p- π conjugation of the ring-closed structure by the oxygen bridge, and can be quantitatively verified by the orbital delocalization index (ODI). The greater the conjugation degree, the smaller the ODI. The ODI of the LUMO of CCO- α is 7.61, smaller than that of CBP- α (10.04), validating the better conjugation of CCO- α (**Fig. R4**), demonstrating the higher conjugation of CCO- α .

Fig. R2. Molecular structure and atomic number of CBP-1 and CCO-1.

Table R2. Charge populations of the donors of CBP-1 and CCO-1.

atom of CBP-1	charge population	atom of CCO-1	charge population
28(N)	0.02007334	29(N)	0.02343438

29(C)	0.02113057
30(C)	-0.04465783
31(C)	0.07025181
32(C)	-0.04465904
33(C)	0.01998657
34(C)	-0.12637609
35(C)	-0.14024945
36(C)	-0.12601293
37(C)	-0.15830304
38(C)	-0.1588223
39(C)	-0.12718455
40(C)	-0.13986429
41(C)	-0.12568424
42(C)	-0.04372426
43(C)	-0.04382026
44(C)	0.08781077
45(C)	-0.1575607
46(C)	-0.13205463
47(C)	-0.13614815
48(C)	-0.12219327
49(C)	-0.12165731
50(C)	-0.13643388
51(C)	-0.13229252
52(C)	-0.15721244
53(C)	0.08887591
54(O)	-0.12699113

30(C)	0.02055087
31(C)	-0.04556518
32(C)	0.07023058
33(C)	-0.04565666
34(C)	0.02123082
35(C)	-0.12632126
36(C)	-0.1385355
37(C)	-0.12567768
38(C)	-0.15825796
39(C)	-0.15800086
40(C)	-0.12542648
41(C)	-0.13885018
42(C)	-0.12661948
43(C)	-0.04292132
44(C)	-0.04315343
45(C)	-0.12309457
46(C)	-0.13706491
47(C)	-0.13160164
48(C)	-0.15802749
49(C)	0.08717482
50(C)	0.08913599
51(C)	-0.1577898
52(C)	-0.13207039
53(C)	-0.13703773
54(C)	-0.12242072
55(O)	-0.12408686

Fig. R3. Schematic illustration of perturbation molecular orbital theory with second-order perturbation and the expression of it. E_{HOMO} and E_{LUMO} are the energy levels of donor and acceptor segments, respectively. E'_{HOMO} and E'_{LUMO} are the HOMO and LUMO energy levels of A-D-type molecules.

$c_{HOMO,r}$ and $c_{LUMO,s}$ are the coefficient of atomic orbitals at the connection point (r and s) for the HOMO and LUMO, respectively. β_{rs} is the exchange integral between atoms r and s . q_r and q_s are the formal charges of atoms r and s , and R_{rs} is the distance between them. ϵ is the permittivity of the reaction medium.

Fig. R4. Calculated HOMO and LUMO distributions and orbital delocalization index (ODI) of LUMOs.

3) In page 4, “Due to weakened D-A interaction and thus lowered CT effect, the LE feature of CCO-1 is enhanced accordingly, indicating the overlap integral between transition orbitals will increase”. I guess the authors mentioned this for the S1 state because this is absolutely not the case for T1, it is rather the opposite. The authors did not discuss why there is a drastic decrease in the overlap factor for the T1 state while this is very much crucial for the rest of the discussion in the paper (e.g. RISC rate). Overall, I think this paragraph is confusing and is missing some analysis.

Response: We thank reviewer’s good suggestion, and we have revised the relevant statements in revised text, “The calculated T₁ energy level of CBP-1 is much lower than that of CCO-1, and thus the theoretical ΔE_{st} of CBP-1 (0.17 eV) becomes larger than that of CCO-1 (0.08 eV). The lower T₁ energy level of CBP-1 discloses a larger exchange integral (K), which can endow the T₁ state of CBP-1 with increased LE component. The $OI_{H/L}$ for the T₁ state of CBP-1 is calculated to be 0.615, larger than that of CCO-1 (0.486), which also validates a higher LE component of the T₁ state.”

4) In page 7, the evolution of the emission energy as a function of the %wt makes a reference to polarization effects but it remains quite speculative in my opinion. A qualitative analysis in terms of the dipole of the molecules could further rationalize this effect.

Response: We thank reviewer’s good suggestion. Molecular polarity index (MPI) is actually emerging as a very useful parameter to study molecular polarity (*Joule* **6**, 1–15 (2022); *ACS Catal.* **12**, 13921–13929 (2021); *ACS Sustainable Chem. Eng.* **9**, 13021–13032 (2021); *Phys. Chem. Chem. Phys.* **23**, 22629–22639 (2021); *J. Phys. Chem. C* **125**, 16378–16390 (2021); *Carbon* **171**, 514–523 (2021)). It has been reported that the dipole moment has a strong dependence on molecular configuration, as exemplified by oxalic acid in **Fig. R5**. In order to obtain an accurate value of dipole moment, it is necessary to add the diffusion function to the base set for recalculation. In this work, we choose MPI to represent the polarity of molecules. The values of MPI are listed in Table S5 in Supplementary

Information.

Fig. R5. Take oxalic acid as an example to calculate its dipole moment and molecular polarity index.

5) In page 7, about the analysis of the transient absorption, I think the analysis is quite misleading and I think that the experiments should be redone. It is quite funny to see two emission peaks at the early stage (below 100 ns) at 440 and 515 nm which do not match at all with the emission energy coming from steady state PL emission. Why would in any ways the 515 nm emission be the delayed fluorescence component? Usually, prompt and delayed fluorescence match quite well and the emission at a later stage around 470 nm matches quite well with the expected emission energy. I clearly doubt that the 515 nm is the delayed fluorescence. The interpretation of the TA is highly questionable, I believe 440 and 515 are actually the LE emissions of the donor and the acceptor (either one or the other) and not the one of the CT band. In my view, the reason is found in the excitation wavelength which is well above the CT-like absorption band edge. Clearly, a higher-lying excited state is probed at this excitation wavelength leading to a path on the manifold of excited states that could be very different compared to the one we would obtain when exciting the CT band.

Response: We thank reviewer's good suggestion. For the doubt whether the two emission peaks at the early stage (below 100 ns) should be attributed to LE emissions or CT emissions, we have tested the transient absorption (TA) spectra and steady state emission spectra of CCO- α (the acceptor of CCO-1), CBP- α (the acceptor of CBP-1) and SXAC (the donor of molecules) with excitation wavelength at 355 nm, as well as the steady state emission spectra of CCO-1 and CBP-1 with excitation wavelengths at 355 nm and 400 nm. As shown in **Fig. R6** and **R7**, from 400 to 600 nm, there are no obvious emission peaks of CCO- α , CBP- α and SXAC. Hence, the stimulated emission signal at 440 and 515 nm could not be attributed to the LE emissions. In addition, the emission spectra with excitation wavelengths at 355 nm and 400 nm of CCO-1 and CBP-1 are similar. Thereby, the TA spectra measured with excitation wavelength at 355 nm are reasonable, and there are also precedents, in which the TA spectra of D-A-type TADF molecules with an emission peak at 486 nm had been tested with an excitation wavelength at 355 nm (*Nat. Photon.* **14**, 636–642 (2020)). After all, according to Kasha's rule, the emission is generated from the radiative decay of the lowest excited state. Furthermore, due to the limitation of testing equipment, we can only measure TA spectra at 355 nm and 410–710 nm excitation wavelengths, and we have attempted to collect the TA spectra excited at 410 nm, but failed to get a spectrogram for analysis because of too weak absorption intensity of the molecules excited at 410 nm.

Fig. R6. PL spectra of (a) CCO-1 and CBP-1 in toluene solutions (10^{-5} M) with excitation wavelengths at 355 and 400 nm, respectively, and (b) CCO- α , CBP- α and SXAC in toluene solutions (10^{-5} M) with excitation wavelengths at 355 nm.

Fig. R7. Transient absorption spectra of CCO- α , CBP- α and SXAC on nanosecond timescales in oxygen-free toluene solutions (10^{-5} M). Excitation wavelength: 355 nm.

6) In page 11, I don't understand why in the hyperfluorescence OLED device with DABNA-NP-TB, the emission comes from DABNA-NP-TB. In view of the electronic levels diagram, it is suggested that a lower band gap is measured for CCO-2 which would suggest that energy transfer takes place from DABNA-NP-TB to CCO-2.

Response: We thank reviewer's good suggestion. We are so sorry that there is something wrong with the electronic levels diagram. The HOMO and LUMO energy levels of DABNA-NP-TB should be -5.09 and -2.36 eV, not -5.36 and -2.09 eV. According to references (*Org. Electron.* **97**, 106275 (2021); *Adv. Funct. Mater.* **31**, 2102017 (2021)), the HOMO energy levels (E_{HOMO}) of DABNA-NP-TB and BN2 were measured by cyclic voltammetry (CV), and the LUMO energy levels (E_{LUMO}) were calculated by E_{HOMO} plus optical band gap (E_{opt} , $E_{\text{LUMO}} = E_{\text{HOMO}} + E_{\text{opt}}$), while in our work, the E_{LUMO} is measured by CV method. To maintain consistency of the calculation method of energy levels, we recalculated the E_{LUMO} of CCO-2 (-2.51 eV) by their method in Fig. 5 in the revised text.

Minor issues:

1) In page 2, it is a minor point I doubt we have a lot of red emitting TADF materials.

Response: We thank reviewer's good suggestion. In introduction, we mainly focus on the research situation of D-A type TADF materials. We have searched the reports of red TADF materials in last two years and found that there are at least 20 red TADF materials with good EL performances, more than

reported efficient blue and deep-blue TADF materials. Some of references are listed as follows:

1. Chen, J-X. et al, Managing Locally Excited and Charge-Transfer Triplet States to Facilitate Up-Conversion in Red TADF Emitters That Are Available for Both Vacuum-and Solution-Processes. *Angew. Chem. Int. Ed.* **60**, 2478–2484 (2021).
 2. Cai, X. et al, Solution-Processable Pure-Red Multiple Resonance-induced Thermally Activated Delayed Fluorescence Emitter for Organic Light-Emitting Diode with External Quantum Efficiency over 20%. *Angew. Chem. Int. Ed.* 10.1002/anie.202216473.
 3. Wu, S. et al, Highly Efficient Green and Red Narrowband Emissive Organic Light-Emitting Diodes Employing Multi-Resonant Thermally Activated Delayed Fluorescence Emitters. *Angew. Chem. Int. Ed.* **61**, e202213697 (2022).
 4. Zhang, H-Y. et al, A Novel Orange-Red Thermally Activated Delayed Fluorescence Emitter with High Molecular Rigidity and Planarity Realizing 32.5% External Quantum Efficiency in Organic Light-Emitting Diodes. *Mater. Horiz.* **9**, 2425–2432 (2022).
 5. Chen, J-X. et al, Optimizing Intermolecular Interactions and Energy Level Alignments of Red TADF Emitters for High-Performance Organic Light-Emitting Diodes. *Small* **18**, 2201548 (2022).
 6. Au-Yeung, C. C. et al, Molecular Design of Efficient Yellow- to Red- Emissive Alkynylgold(III) Complexes for the Realization of Thermally Activated Delayed Fluorescence (TADF) and Their Applications in Solution-Processed Organic Light-Emitting Devices. *Chem. Sci.* **12**, 9516–9527 (2021).
 7. Liu, Z. et al, Efficient Intramolecular Charge-Transfer Fluorophores Based on Substituted Triphenylphosphine Donors. *Angew. Chem. Int. Ed.* **60**,15049–15053 (2021).
 8. Wang, Y-Y. et al, Positive Impact of Chromophore Flexibility on the Efficiency of Red Thermally Activated Delayed Fluorescence Materials. *Mater. Horiz.* **8**, 1297–1303 (2021).
 9. Li, Z. et al, Optimizing Charge Transfer and Out-Coupling of A Quasi-Planar Deep-Red TADF Emitter: towards Rec.2020 Gamut and External Quantum Efficiency beyond 30 %. *Angew. Chem. Int. Ed.* **60**, 14846–14851 (2021).
 10. Zeng, X. et al, Nitrogen-Embedded Multi-Resonance Heteroaromatics with Prolonged Homogeneous Hexatomic Rings. *Angew. Chem. Int. Ed.* **61**, e20211718 (2022).
 11. He, J-L. et al, An Extended π -Backbone for Highly Efficient Near-Infrared Thermally Activated Delayed Fluorescence with Enhanced Horizontal Molecular Orientation. *Mater. Horiz.* **9**, 772–779 (2022).
- 2) In page 3, “And the root of the mean of squared displacement (RMSD)” should be replaced by “And the root mean squared displacement (RMSD)”.

Response: We thank reviewer’s good suggestion, and we have made the correction accordingly in the revised text.

Reviewer #2:

1. Discussion based on the energy gaps are not sufficient for the RISC process. It would be better to conduct SOC calculations to clarify effect of the bridging oxygen atom.

Response: We thank reviewer’s good suggestion, and have calculated the SOC values of CBP-1 and CCO-1 for comparison and discussion. It is found that, due to more different transition characteristics between S_1 and T_1 states, the SOC constant of CBP-1 (0.77 cm^{-1}) is larger than that of CCO-1 (0.36

cm^{-1}), indicating the bridging oxygen atom does not contribute to SOC. But the ring-closed structure by oxygen atom can influence excited-state energy levels, leading to a smaller ΔE_{st} for CCO-1. So, the k_{RISC} from T_1 to S_1 state of CCO-1 is calculated to be $2.46 \times 10^6 \text{ s}^{-1}$, larger than that of CBP-1 ($9.14 \times 10^5 \text{ s}^{-1}$), despite the smaller SOC constant. We have added the relevant analysis and discussion of SOC in the revised text.

2. There are many reports of blue and green HF devices using MR-TADF materials. EQE and lifetime should be compared with those reported for the HF devices as for the simple TADF devices (Table S7).

Response: We thank reviewer's good suggestion. Since the works measuring the device operational lifetimes of MR-TADF molecules are limited, we have summarized the works using sensitizers for MR-TADF materials and collected the works with traditional hosts for MR-TADF materials. And we have added the comparison of EQEs and lifetimes of blue and green HF devices in Table S10.

3. The operational lifetime of HF3 has only been measured partway through.

Response: We thank reviewer's good suggestion. We believe that there is still room for further optimization of previous device structure. Herein, we have designed supplementary device structures and obtained better results. The latest results have been updated in revised text.

4. Given the measurement error and standard deviation between devices, there is no scientific basis for 4 significant digits for EQEs.

Response: We thank reviewer's good suggestion, and we have corrected the significant digits for EQEs as 3 in the revised text.

Thank you for your review and kind consideration. We look forward to hearing from you.

Reviewers' Comments:

Reviewer #1:

Remarks to the Author:

My report is attached below

I read the article and the answer to my comments. I think some of the changes are satisfying but clearly two following ones are not:

2) In page 4, the interpretation of the orbital evolution going from CBP-1 to CCO-1 is misleading. They mentioned increase in conjugation to explain the decrease of the LUMO going from CBP-1 to CCO-1 is not correct, adding an oxygen to planarize the acceptor does not result in an increase in conjugation. Regarding the evolution of the HOMO energy, I clearly do not know what they mean by CT effect. Evolution of the orbital energies should be discussed in terms of inductive and/or mesomeric electron donating or withdrawing effects and I bet the effect is mainly inductive in view of the symmetric stabilization of the HOMO and LUMO orbitals. Clearly, this paragraph has to be revised.

Response: We thank reviewer's good suggestion. Well, the inductive effect is a short-range force, and gets very weak at the third carbon. We compare the charge distributions of CBP-1 and CCO-1, and find that they have similar distributions in donors, which are listed in **Fig. R2** and **Table R2**. Hence, the evolution of HOMO energy level should not be ascribed to the inductive effect at least. Regarding the weakened D-A interaction mentioned in manuscript, it can be explained by perturbation molecular orbital theory. The weak D-A interaction leads to a weak electrostatic interaction, which would increase the energy gap between the frontier molecular orbital, manifesting as the increase of LUMO energy level (E_{LUMO}) and the decrease of HOMO energy level (E_{HOMO}) (**Fig. R3**). Indeed, the E_{HOMO} of CCO-1 is decreased by 0.058 eV relative to that of CBP-1, while the E_{LUMO} of CCO-1 is also reduced by 0.066 eV instead of being increased owing to improved conjugation of CCO- α segment, which is induced by the enhanced molecular planarity and the formation of p- π conjugation of the ring-closed

Answer: I think this analysis does not answer the evolution from CBP-1 to CCO-1. I would like to remind the authors a few facts:

- Conjugation is the overlap of one p-orbital with another across an adjacent σ bond. This definition does not distinguish between the HOMO and LUMO orbitals. Increase in conjugation leads to the stabilization of the LUMO and the destabilization of the HOMO. The authors discussed this delocalization effect only on the LUMO. Why not consider such an effect on the HOMO?
- The definition of the ODI index is not reported either in the paper or in the SI. No reference to the ODI index appears in the bibliography.
- I really don't understand the sketches in Fig S10:
 - o About the covalent interaction: The position of the HOMO (LUMO) in the D-A compound is determined by the electronic interaction between the HOMOs of the D and the A units, not by the interaction between the HOMO and the LUMO
 - o About the electrostatic effect: this looks misleading to me. The authors are mixing "classical" arguments with "quantum" ones. I really don't understand.
- About inductive effects: I really urge the authors to read these two papers [10.1063/1.2713096](https://doi.org/10.1063/1.2713096) and [10.1002/cphc.200800122](https://doi.org/10.1002/cphc.200800122) where inductive effects and mesomeric effects are discussed extensively. Again, inductive effects lead to quasi symmetric (de)stabilization of the HOMO and LUMO orbitals. This is a more reasonable and simple explanation than what is proposed.

5) In page 7, about the analysis of the transient absorption, I think the analysis is quite misleading and I think that the experiments should be redone. It is quite funny to see two emission peaks at the early stage (below 100 ns) at 440 and 515 nm which do not match at all with the emission energy

coming from steady state PL emission. Why would in any ways the 515 nm emission be the delayed fluorescence component? Usually, prompt and delayed fluorescence match quite well and the emission at a later stage around 470 nm matches quite well with the expected emission energy. I clearly doubt that the 515 nm is the delayed fluorescence. The interpretation of the TA is highly questionable, I believe 440 and 515 are actually the LE emissions of the donor and the acceptor (either one or the other) and not the one of the CT band. In my view, the reason is found in the excitation wavelength which is well above the CT-like absorption band edge. Clearly, a higher-lying excited state is probed at this excitation wavelength leading to a path on the manifold of excited states that could be very different compared to the one we would obtain when exciting the CT band.

Response: We thank reviewer's good suggestion. For the doubt whether the two emission peaks at the early stage (below 100 ns) should be attributed to LE emissions or CT emissions, we have tested the transient absorption (TA) spectra and steady state emission spectra of CCO- α (the acceptor of CCO 1), CBP- α (the acceptor of CBP-1) and SXAC (the donor of molecules) with excitation wavelength at 355 nm, as well as the steady state emission spectra of CCO-1 and CBP-1 with excitation wavelengths at 355 nm and 400 nm. As shown in **Fig. R6** and **R7**, from 400 to 600 nm, there are no obvious emission peaks of CCO- α , CBP- α and SXAC. Hence, the stimulated emission signal at 440 and 515 nm could not be attributed to the LE emissions. In addition, the emission spectra with excitation wavelengths at 355 nm and 400 nm of CCO-1 and CBP-1 are similar. Thereby, the TA spectra measured with excitation wavelength at 355 nm are reasonable, and there are also precedents, in which the TA spectra of D-A type TADF molecules with an emission peak at 486 nm had been tested with an excitation wavelength at 355 nm (*Nat. Photon.* **14**, 636–642 (2020)). After all, according to Kasha's rule, the emission is generated from the radiative decay of the lowest excited state. Furthermore, due to the limitation of testing equipment, we can only measure TA spectra at 355 nm and 410–710 nm excitation wavelengths, and we have attempted to collect the TA spectra excited at 410 nm, but failed to get a spectrogram for analysis because of too weak absorption intensity of the molecules excited at 410 nm.

Answer: I appreciate the further experiments carrier out by the authors which show that the peaks at 440 and 515 are not due to LE emission from the D and the A units. Still, the discrepancies between the TA of CCO-1 in the 30-100 ns range and the 110-1000 ns range are very different. It does not seem to be associated to the ground state bleaching and clearly these two bands do not match with the PL spectrum. So the origin of these two is not clear. The additional text in the manuscript is not satisfying.

Reviewer #2:

Remarks to the Author:

The revised manuscript has addressed my original concerns, but lacks the details of the description below. It is necessary to provide details, including computational methods, data, and how to estimate k_{RISC} values from SOC constant and ST-gap in the supporting information. Additionally, in its current form, most results lack the computational methods and Cartesian coordinates required for reproduction and publication.

Original comment: 1. Discussion based on the energy gaps are not sufficient for the RISC process. It would be better to conduct SOC calculations to clarify effect of the bridging oxygen atom.

Response: We thank reviewer's good suggestion, and have calculated the SOC values of CBP-1 and CCO-1 for comparison and discussion. It is found that, due to more different transition characteristics between S1 and T1 states, the SOC constant of CBP-1 (0.77 cm^{-1}) is larger than that of CCO-1 (0.36 cm^{-1}), indicating the bridging oxygen atom does not contribute to SOC. But the ring-closed structure by oxygen atom can influence excited-state energy levels, leading to a smaller ΔE_{ST} for CCO-1. So, the k_{RISC} from T1 to S1 state of CCO-1 is calculated to be $2.46 \times 10^6 \text{ s}^{-1}$, larger than that of CBP-1 ($9.14 \times 10^5 \text{ s}^{-1}$), despite the smaller SOC constant. We have added the relevant analysis and discussion of SOC in the revised text.

1 **Reviewer #1**

2 1. Conjugation is the overlap of one p-orbital with another across an adjacent σ bond. This definition
3 does not distinguish between the HOMO and LUMO orbitals. Increase in conjugation leads to the
4 stabilization of the LUMO and the destabilization of the HOMO. The authors discussed this
5 delocalization effect only on the LUMO. Why not consider such an effect on the HOMO?

6
7 Response: We thank reviewer's good suggestion. Unlike conventional well-conjugated planar
8 molecules such as pentacene and oligothiophene, D-A-type TADF materials have highly twisted
9 structures with very large dihedral angles almost approaching 90° between donor and acceptor
10 segments, which lead to poor conjugation between donor and acceptor segments. Accordingly, their
11 HOMOs mainly focus on the segments which have strong electron-donating ability, namely the donors.
12 For the TADF materials studied in this work, they have identical donor segments. Hence, we primarily
13 discuss the influence of the delocalization effect on the LUMO rather than the HOMO. Due to the
14 highly twisted structures and thus fully separated HOMOs and LUMOs of these TADF materials, the
15 influence of delocalization effect on the HOMO is actually ignorable.

16
17 2. The definition of the ODI index is not reported either in the paper or in the SI. No reference to the
18 ODI index appears in the bibliography.

19
20 Response: We thank reviewer's good suggestion. The ODI index of a certain orbit i can be calculated
21 by the following formula (*Multiwfn Manual Ch. 4* available at <http://sobereva.com/multiwfn>):

$$22 \quad \text{ODI}_i = 0.01 \times \sum_A (\Theta_{A,i})^2$$

23 where, $\Theta_{A,i}$ is the composition of atom A in orbital i . The lower the ODI, the stronger the orbital
24 delocalization. The relevant information of ODI has been added in the revised method.

25 3. I really don't understand the sketches in Fig S10:

26 (1) About the covalent interaction: The position of the HOMO (LUMO) in the D-A compound is
27 determined by the electronic interaction between the HOMOs of the D and the A units, not by the
28 interaction between the HOMO and the LUMO.

29 (2) About the electrostatic effect: this looks misleading to me. The authors are mixing "classical"
30 arguments with "quantum" ones. I really don't understand.

31
32 Response: We thank reviewer's good questions. Fig S10 is the schematic illustration of generalized
33 perturbation molecular orbital theory with second-order perturbation, in which the "perturbation
34 molecular orbital theory" should be clarified with a qualifier and corrected as "generalized perturbation
35 molecular orbital theory". The generalized perturbation theory addresses itself to the problem of what
36 happens to the energy when two reagents interact, and deals with the process of electron transfer during
37 bond formation. We analyzed the evolution of orbital energy levels based on this theory, and have
38 made revisions in the revised text. According to the references, the overlap between atomic orbitals of
39 two molecules (considered as one composite system) is introduced in the theory, and the explicit
40 expressions for the interaction energy will be required to represent the atom-atom or orbital-orbital
41 interactions if large charge-charge interactions are present. The energy of molecular orbital of one

42 molecule after perturbation is the result of the combined influence of orbitals of another one. For the
43 system with donor-acceptor interaction, the occupied orbitals of electron-donating molecules mainly
44 interact with the unoccupied orbitals of electron-withdrawing ones. Besides, according to frontier
45 molecular orbital theory, the most active orbitals are the frontier molecular orbitals, that is HOMO and
46 LUMO. We hope these introductions could help you to better understand Fig. S10. For details of this
47 theory, please refer to the following references: *J. Am. Chem. Soc.* **90**, 543–552 (1968); *Chemical*
48 *reactivity and reaction paths Ch.4* (John Wiley & Sons, New York, 1974).

49

50 4. About inductive effects: I really urge the authors to read these two papers 10.1063/1.2713096 and
51 10.1002/cphc.200800122 where inductive effects and mesomeric effects are discussed extensively.
52 Again, inductive effects lead to quasi symmetric (de)stabilization of the HOMO and LUMO orbitals.
53 This is a more reasonable and simple explanation than what is proposed.

54

55 Response: We thank reviewer's good suggestion. We have read these two papers carefully, and gained
56 a lot of inspiration. It is a good idea to regulate the frontier orbital energy level of conjugated molecules
57 by the groups with different inductive effects and mesomeric effects based on the LCAO coefficients,
58 which can provide helpful clues for designing new efficient molecules in the future. In this work, the
59 charge distributions on the donor segments of CCO-1 and CBP-0 are almost identical (Fig. R1 and
60 Table R1), indicating the inductive effect hardly influence the energy levels of HOMOs. This is
61 probably because the bridging oxygen atom is located at the *meta*-position to the donor segment and
62 thus the inductive effect of bridging oxygen atom for donor segment is actually very weak. But the
63 bridging oxygen atom located at the *ortho*-position to carbonyl group may exert an inductive effect
64 over acceptor and may contribute to the decrease of LUMO energy level to some extent. We have made
65 revisions in the statement in the revised text, as "the decreased LUMO energy level is mainly ascribed
66 to the improved conjugation of CCO- α moiety, and the inductive effect of the oxygen bridge may also
67 have some contributions (*ChemPhysChem* **9**, 1519–1523 (2008); *J. Chem. Phys.* **126**, 111101 (2007))".
68 The mentioned references are cited as well.

69

71

Fig. R1. Molecular structure and atomic number of CBP-1 and CCO-1.

72

73

Table R1. Charge populations of the donors of CBP-1 and CCO-1.

atom of CBP-1	charge population	atom of CCO-1	charge population
28(N)	0.02007334	29(N)	0.02343438
29(C)	0.02113057	30(C)	0.02055087
30(C)	-0.04465783	31(C)	-0.04556518

31(C)	0.07025181
32(C)	-0.04465904
33(C)	0.01998657
34(C)	-0.12637609
35(C)	-0.14024945
36(C)	-0.12601293
37(C)	-0.15830304
38(C)	-0.1588223
39(C)	-0.12718455
40(C)	-0.13986429
41(C)	-0.12568424
42(C)	-0.04372426
43(C)	-0.04382026
44(C)	0.08781077
45(C)	-0.1575607
46(C)	-0.13205463
47(C)	-0.13614815
48(C)	-0.12219327
49(C)	-0.12165731
50(C)	-0.13643388
51(C)	-0.13229252
52(C)	-0.15721244
53(C)	0.08887591
54(O)	-0.12699113

32(C)	0.07023058
33(C)	-0.04565666
34(C)	0.02123082
35(C)	-0.12632126
36(C)	-0.1385355
37(C)	-0.12567768
38(C)	-0.15825796
39(C)	-0.15800086
40(C)	-0.12542648
41(C)	-0.13885018
42(C)	-0.12661948
43(C)	-0.04292132
44(C)	-0.04315343
45(C)	-0.12309457
46(C)	-0.13706491
47(C)	-0.13160164
48(C)	-0.15802749
49(C)	0.08717482
50(C)	0.08913599
51(C)	-0.1577898
52(C)	-0.13207039
53(C)	-0.13703773
54(C)	-0.12242072
55(O)	-0.12408686

74

75 5. Answer: I appreciate the further experiments carrier out by the authors which show that the peaks at
76 440 and 515 are not due to LE emission from the D and the A units. Still, the discrepancies between
77 the TA of CCO-1 in the 30-100 ns range and the 110-1000 ns range are very different. It does not seem
78 to be associated to the ground state bleaching and clearly these two bands do not match with the PL
79 spectrum. So the origin of these two is not clear. The additional text in the manuscript is not satisfying.

80

81 Response: We thank reviewer's good suggestion. We have added the explanation of the temporal
82 dynamics feature of prompt and delayed fluorescence peaks of TA spectra after Supplementary Fig. 25
83 in Supplementary Information, described as follows. From 30 to 100 ns, the prompt fluorescence peak
84 is gradually red-shifted from 432 to 463 nm, while the delayed ones exhibit an opposite trend. The
85 temporal dynamics feature can be ascribed to the various D-A twisting configuration, and thus different
86 CT feature. The initial prompt fluorescence generates by conformers with weak CT feature, along with
87 fast k_r and large S_1 energy, and then the conformers with relatively strong CT feature emit, leading to
88 the red-shifted emission. In the case of delayed fluorescence, the S_1 configuration with relatively low
89 energy undergo RISC process from T_1 state firstly, owing to their smaller ΔE_{st} , and emit fluorescence,
90 followed by other configurations with relatively high energy, which lead to the blue-shift (*J. Phys.*
91 *Chem. Lett.* **13**, 1839–1844 (2022)). On the other hand, it is worth noting that before 30 ns, there is

92 only one peak located at ~440 nm (Supplementary Fig. 23). The peak located at 515 nm appears after
93 30 ns owing to the slower rate of RISC process than that of radiative decay process. We hope the above
94 description will be satisfactory to you.

95

96 **Reviewer #2**

97 The revised manuscript has addressed my original concerns, but lacks the details of the description
98 below. It is necessary to provide details, including computational methods, data, and how to estimate
99 kRISC values from SOC constant and ST-gap in the supporting information. Additionally, in its current
100 form, most results lack the computational methods and Cartesian coordinates required for reproduction
101 and publication.

102

103 Response: We thank reviewer's good suggestion. The computational methods and the approach to
104 estimate kRISC values from SOC constant and ST-gap have been added in the revised method section,
105 and the Cartesian coordinates have been added in the additional excel for raw data.

106

REVIEWER COMMENTS

Reviewer #1 (Remarks to the Author):

See the attached review

I am still very much puzzled by the rationale provided to explain the evolution of the energies of the frontier molecular orbitals (HOMO and LUMO) when going from CPB-1 to CCO-1. The authors provided an explanation in terms of second-order perturbation molecular orbital theory considering that the energy of the HOMO (LUMO) is modulated by the electronic interaction between the HOMO and the LUMO plus a classical electrostatic term. This is actually not right, the energy of the HOMO (LUMO) of a donor-acceptor material is driven in a first approximation by both the relative energy and the electronic interaction between the HOMO (LUMO) of the electron-donating unit and the HOMO (LUMO) of the electron-accepting unit.

To prove our hypothesis, we carried out DFT calculations using the PBE0 functional and the 6-31G(d,p) basis on the electron-donating and -accepting units and the donor-acceptor materials.

Here is the orbital plot for CBP-1:

Here is the orbital plot for CCO-1:

The electronic interaction between the HOMOs (LUMOs) of the electron-accepting and the electron-donating units is rather similar in view of the similar dihedral angle between the electron-donating and -accepting units in CBP-1 and CCO-1 (see Figure below). Thus, the energy of the HOMO (LUMO) of the donor-acceptor materials is determined by the relative energy between the HOMOs (LUMOs) of the electron-donating and the electron-accepting unit.

Strikingly, the LUMO of CCO-1 is lower in energy than the LUMO of CBP-1 in line with the relative energy between the LUMOs of the electron-accepting units, so that the difference in energy between the LUMOs of the electron-accepting unit in CBP-1 and in CCO-1 qualitatively explains the larger stabilization of the LUMOs of CBP-1 with respect to CCO-1.

An identical reasoning could be adopted for the HOMO of CBP-1 and CCO-1. The HOMO energy of the electron-donating unit in CBP-1 is higher than in CCO-1 allowing for higher destabilization of the HOMO of CBP-1 with respect to CCO-1 by electronic interaction between the HOMOs of the electron-donating and -accepting units.

With these simple interaction arguments between frontier molecular orbitals, it is rather straightforward to rationalize the frontier orbital energies evolution from CBP-1 to CCO-1. In this case, it is essentially driven by the energies of the frontier orbitals of the electron-accepting units.

The followings are our point-by-point responses to the reviewers' comments and suggestions.

Reviewer #1 comments

I am still very much puzzled by the rationale provided to explain the evolution of the energies of the frontier molecular orbitals (HOMO and LUMO) when going from CPB-1 to CCO-1. The authors provided an explanation in terms of second-order perturbation molecular orbital theory considering that the energy of the HOMO (LUMO) is modulated by the electronic interaction between the HOMO and the LUMO plus a classical electrostatic term. This is actually not right, the energy of the HOMO (LUMO) of a donor-acceptor material is driven in a first approximation by both the relative energy and the electronic interaction between the HOMO (LUMO) of the electron-donating unit and the HOMO (LUMO) of the electron-accepting unit.

To prove our hypothesis, we carried out DFT calculations using the PBE0 functional and the 6-31G(d,p) basis on the electron-donating and -accepting units and the donor-acceptor materials.

Here is the orbital plot for CBP-1:

Here is the orbital plot for CCO-1:

The electronic interaction between the HOMOs (LUMOs) of the electron-accepting and the electron-donating units is rather similar in view of the similar dihedral angle between the electron-donating and -accepting units in CBP-1 and CCO-1 (see Figure below). Thus, the energy of the HOMO (LUMO) of the donor-acceptor materials is determined by the relative energy between the HOMOs (LUMOs) of the electron-

donating and the electron-accepting unit.

Strikingly, the LUMO of CCO-1 is lower in energy than the LUMO of CBP-1 in line with the relative energy between the LUMOs of the electron-accepting units, so that the difference in energy between the LUMOs of the electron-accepting unit in CBP-1 and in CCO-1 qualitatively explains the larger stabilization of the LUMOs of CBP-1 with respect to CCO-1.

An identical reasoning could be adopted for the HOMO of CBP-1 and CCO-1. The HOMO energy of the electron-donating unit in CBP-1 is higher than in CCO-1 allowing for higher destabilization of the HOMO of CBP-1 with respect to CCO-1 by electronic interaction between the HOMOs of the electron-donating and -accepting units.

With these simple interaction arguments between frontier molecular orbitals, it is rather straightforward to rationalize the frontier orbital energies evolution from CBP-1 to CCO-1. In this case, it is essentially driven by the energies of the frontier orbitals of the electron-accepting units.

Response to reviewer comments

We thank reviewer's detailed explanation and guidance, and we very much appreciate your point of view, which has greatly inspired us. We have given up the explanation in terms of second-order perturbation molecular orbital theory, and have revised the explanation of the evolution of energy levels of the frontier molecular orbitals (HOMO and LUMO) according to the suggestions. And the cause of the different relative energies between the HOMOs (LUMOs) of CCO- α and CBP- α are explained by referring to *ChemPhysChem* **9**, 1519–1523 (2008) and *J. Chem. Phys.* **126**, 111101 (2007). The following is the revised explanation.

The energy level of the HOMO (LUMO) of a D-A-type molecule is driven in a first approximation by both the relative energy and the electronic interaction between the HOMOs (LUMOs) of the electron donor (D) and acceptor (A) fragments. With the inductive effect of the oxygen bridge in CCO- α , the LUMO energy level (E_{LUMO}) of CCO- α is lowered than that of CBP- α . In addition, the electronic interaction of CCO-1 and CBP-1 is similar due to the close dihedral angles between D and A fragments.

Hence, the lower relative energy of CCO- α with respect to CBP- α results in a lower E_{LUMO} of CCO-1. An identical reasoning can be adopted for the evolution of HOMO energy level (E_{HOMO}) of CBP-1 and CCO-1.

However, owing to the small orbital coefficients in HOMO at the C₆ and C₂₆ positions of CBP-1, the influence of the inductive effect on E_{HOMO} is weaker in comparison with that on E_{HOMO} . Besides, the E_{HOMO} of CCO- α is further destabilized by the mesomeric effect of the oxygen bridge, due to the low-lying energy level of the oxygen's lone pair in relation to this orbital. Thus, the decrease in E_{HOMO} of CCO- α is smaller than that of E_{LUMO} . Overall, the HOMO-LUMO gap of CCO-1 (3.68 eV) is decreased compared with that of CBP-1 (3.69 eV), but the difference between them is actually neglectable.

Fig. R1 The energy level and orbital distribution of CBP-1, CCO-1, CBP- α , CCO- α and SXAC.

Fig. R2 The dihedral angles in CBP-1 and CCO-1.

Thank you again for your seriousness and responsibility. We look forward to hearing from you soon.

Reviewers' Comments:

Reviewer #1:

Remarks to the Author:

The authors of the paper have answered all my comments.
I don't see any additional points that would need a revision